# Interpolating Convex and Non-Convex Tensor Decompositions via the Subspace Norm

**Qinqing Zheng**
University of Chicago
qinqing@cs.uchicago.edu

**Ryota Tomioka**
Toyota Technological Institute at Chicago
tomioka@ttic.edu

## Abstract

We consider the problem of recovering a low-rank tensor from its noisy observation. Previous work has shown a recovery guarantee with signal to noise ratio $O(n^{\lceil K/2 \rceil /2})$ for recovering a $K$th order rank one tensor of size $n \times \cdots \times n$ by recursive unfolding. In this paper, we first improve this bound to $O(n^{K/4})$ by a much simpler approach, but with a more careful analysis. Then we propose a new norm called the *subspace* norm, which is based on the Kronecker products of factors obtained by the proposed simple estimator. The imposed Kronecker structure allows us to show a nearly ideal $O(\sqrt{n} + \sqrt{H^{K-1}})$ bound, in which the parameter $H$ controls the blend from the non-convex estimator to mode-wise nuclear norm minimization. Furthermore, we empirically demonstrate that the subspace norm achieves the nearly ideal denoising performance even with $H = O(1)$.

## 1 Introduction

Tensor is a natural way to express higher order interactions for a variety of data and tensor decomposition has been successfully applied to wide areas ranging from chemometrics, signal processing, to neuroimaging; see [15, 18] for a survey. Moreover, recently it has become an active area in the context of learning latent variable models [3].

Many problems related to tensors, such as, finding the rank, or a best rank-one approaximation of a tensor is known to be NP hard [11, 8]. Nevertheless we can address statistical problems, such as, how well we can recover a low-rank tensor from its randomly corrupted version (tensor denoising) or from partial observations (tensor completion). Since we can convert a tensor into a matrix by an operation known as *unfolding*, recent work [25, 19, 20, 13] has shown that we do get nontrivial guarantees by using some norms or singular value decompositions. More specifically, Richard & Montanari [20] has shown that when a rank-one $K$th order tensor of size $n \times \cdots \times n$ is corrupted by standard Gaussian noise, a nontrivial bound can be shown with high probability if the signal to noise ratio $\beta / \sigma \gtrsim n^{\lceil K/2 \rceil /2}$ by a method called the recursive unfolding[1]. Note that $\beta / \sigma \gtrsim \sqrt{n}$ is sufficient for matrices ($K = 2$) and also for tensors if we use the best rank-one approximation (which is known to be NP hard) as an estimator. On the other hand, Jain & Oh [13] analyzed the tensor completion problem and proposed an algorithm that requires $O(n^{3/2} \cdot \mathrm{polylog}(n))$ samples for $K = 3$; while information theoretically we need at least $\Omega(n)$ samples and the intractable maximum likelihood estimator would require $O(n \cdot \mathrm{polylog}(n))$ samples. Therefore, in both settings, there is a wide gap between the ideal estimator and current polynomial time algorithms. A subtle question that we will address in this paper is whether we need to unfold the tensor so that the resulting matrix become as square as possible, which was the reasoning underlying both [19, 20].

As a parallel development, non-convex estimators based on alternating minimization or nonlinear optimization [1, 21] have been widely applied and have performed very well when appropriately

Table 1: Comparison of required signal-to-noise ratio $\beta/\sigma$ of different algorithms for recovering a $K$th order rank one tensor of size $n \times \cdots \times n$ contaminated by Gaussian noise with Standard deviation $\sigma$. See model (2). The bound for the ordinary unfolding is shown in Corollary 1. The bound for the subspace norm is shown in Theorem 2. The ideal estimator is proven in Appendix A.

| Overlapped/ Latent nuclear norm[23] | Recursive unfolding[20]/ square norm[19] | Ordinary un-folding | Subspace norm (proposed) | Ideal |
|---|---|---|---|---|
| $O(n^{(K-1)/2})$ | $O(n^{\lceil K/2 \rceil /2})$ | $\boldsymbol{O(n^{K/4})}$ | $\boldsymbol{O(\sqrt{n} + \sqrt{H^{K-1}})}$ | $\boldsymbol{O(\sqrt{nK\log(K)})}$ |

set up. Therefore it would be of fundamental importance to connect the wisdom of non-convex estimators with the more theoretically motivated estimators that recently emerged.

In this paper, we explore such a connection by defining a new norm based on Kronecker products of factors that can be obtained by simple mode-wise singular value decomposition (SVD) of unfoldings (see notation section below), also known as the higher-order singular value decomposition (HOSVD) [6, 7]. We first study the non-asymptotic behavior of the leading singular vector from the ordinary (rectangular) unfolding $\boldsymbol{X}_{(k)}$ and show a nontrivial bound for signal to noise ratio $\beta/\sigma \gtrsim n^{K/4}$. Thus the result also applies to odd order tensors confirming a conjecture in [20]. Furthermore, this motivates us to use the solution of mode-wise truncated SVDs to construct a new norm. We propose the subspace norm, which predicts an unknown low-rank tensor as a mixture of $K$ low-rank tensors, in which each term takes the form

$$\mathrm{fold}_k(\boldsymbol{M}^{(k)}(\widehat{\boldsymbol{P}}^{(1)} \otimes \cdots \otimes \widehat{\boldsymbol{P}}^{(k-1)} \otimes \widehat{\boldsymbol{P}}^{(k+1)} \otimes \cdots \otimes \widehat{\boldsymbol{P}}^{(K)})^\top),$$

where $\mathrm{fold}_k$ is the inverse of unfolding $(\cdot)_{(k)}$, $\otimes$ denotes the Kronecker product, and $\widehat{\boldsymbol{P}}^{(k)} \in \mathbb{R}^{n \times H}$ is a orthonormal matrix estimated from the mode-$k$ unfolding of the observed tensor, for $k = 1, \ldots, K$; $H$ is a user-defined parameter, and $\boldsymbol{M}^{(k)} \in \mathbb{R}^{n \times H^{K-1}}$. Our theory tells us that with sufficiently high signal-to-noise ratio the estimated $\widehat{\boldsymbol{P}}^{(k)}$ spans the true factors.

We highlight our contributions below:

1. We prove that the required signal-to-noise ratio for recovering a $K$th order rank one tensor from the ordinary unfolding is $O(n^{K/4})$. Our analysis shows a curious two phase behavior: with high probability, when $n^{K/4} \precsim \beta/\sigma \precsim n^{K/2}$, the error shows a fast decay as $1/\beta^4$; for $\beta/\sigma \gtrsim n^{K/2}$, the error decays slowly as $1/\beta^2$. We confirm this in a numerical simulation.
2. The proposed subspace norm is an interpolation between the intractable estimators that directly control the rank (e.g., HOSVD) and the tractable norm-based estimators. It becomes equivalent to the latent trace norm [23] when $H = n$ at the cost of increased signal-to-noise ratio threshold (see Table 1).
3. The proposed estimator is more efficient than previously proposed norm based estimators, because the size of the SVD required in the algorithm is reduced from $n \times n^{K-1}$ to $n \times H^{K-1}$.
4. We also empirically demonstrate that the proposed subspace norm performs nearly optimally for constant order $H$.

**Notation**

Let $\mathcal{X} \in \mathbb{R}^{n_1 \times n_2 \times \cdots \times n_K}$ be a $K$th order tensor. We will often use $n_1 = \cdots = n_K = n$ to simplify the notation but all the results in this paper generalizes to general dimensions. The inner product between a pair of tensors is defined as the inner products of them as vectors; i.e., $\langle \mathcal{X}, \mathcal{W} \rangle = \langle \mathrm{vec}(\mathcal{X}), \mathrm{vec}(\mathcal{W}) \rangle$. For $\boldsymbol{u} \in \mathbb{R}^{n_1}, \boldsymbol{v} \in \mathbb{R}^{n_2}, \boldsymbol{w} \in \mathbb{R}^{n_3}$, $\boldsymbol{u} \circ \boldsymbol{v} \circ \boldsymbol{w}$ denotes the $n_1 \times n_2 \times n_3$ *rank-one* tensor whose $i, j, k$ entry is $u_i v_j w_k$. The rank of $\mathcal{X}$ is the minimum number of rank-one tensors required to write $\mathcal{X}$ as a linear combination of them. A mode-$k$ fiber of tensor $\mathcal{X}$ is an $n_k$ dimensional vector that is obtained by fixing all but the $k$th index of $\mathcal{X}$. The mode-$k$ unfolding $\boldsymbol{X}_{(k)}$ of tensor $\mathcal{X}$ is an $n_k \times \prod_{k' \neq k} n_{k'}$ matrix constructed by concatenating all the mode-$k$ fibers along columns. We denote the spectral and Frobenius norms for matrices by $\| \cdot \|$ and $\| \cdot \|_F$, respectively.

## 2 The power of ordinary unfolding

### 2.1 A perturbation bound for the left singular vector

We first establish a bound on recovering the left singular vector of a rank-one $n \times m$ matrix (with $m > n$) perturbed by random Gaussian noise.

Consider the following model known as the information plus noise model [4]:

$$\tilde{X} = \beta \boldsymbol{u}\boldsymbol{v}^\top + \sigma \boldsymbol{E}, \tag{1}$$

where $\boldsymbol{u}$ and $\boldsymbol{v}$ are unit vectors, $\beta$ is the signal strength, $\sigma$ is the noise standard deviation, and the noise matrix $\boldsymbol{E}$ is assumed to be random with entries sampled i.i.d. from the standard normal distribution. Our goal is to lower-bound the correlation between $\boldsymbol{u}$ and the top left singular vector $\hat{\boldsymbol{u}}$ of $\tilde{X}$ for signal-to-noise ratio $\beta/\sigma \gtrsim (mn)^{1/4}$ with high probability.

A direct application of the classic Wedin perturbation theorem [28] to the rectangular matrix $\tilde{X}$ does not provide us the desired result. This is because it requires the signal to noise ratio $\beta/\sigma \geq 2\|\boldsymbol{E}\|$. Since the spectral norm of $\boldsymbol{E}$ scales as $O_p(\sqrt{n} + \sqrt{m})$ [27], this would mean that we require $\beta/\sigma \gtrsim m^{1/2}$; i.e., the threshold is dominated by the number of columns $m$, if $m \geq n$.

Alternatively, we can view $\hat{\boldsymbol{u}}$ as the leading eigenvector of $\tilde{X}\tilde{X}^\top$, a square matrix. Our key insight is that we can decompose $\tilde{X}\tilde{X}^\top$ as follows:

$$\tilde{X}\tilde{X}^\top = (\beta^2 \boldsymbol{u}\boldsymbol{u}^\top + m\sigma^2 \boldsymbol{I}) + (\sigma^2 \boldsymbol{E}\boldsymbol{E}^\top - m\sigma^2 \boldsymbol{I}) + \beta\sigma(\boldsymbol{u}\boldsymbol{v}^\top \boldsymbol{E}^\top + \boldsymbol{E}\boldsymbol{v}\boldsymbol{u}^\top).$$

Note that $\boldsymbol{u}$ is the leading eigenvector of the first term because adding an identity matrix does not change the eigenvectors. Moreover, we notice that there are two noise terms: the first term is a centered Wishart matrix and it is independent of the signal $\beta$; the second term is Gaussian distributed and depends on the signal $\beta$.

This implies a two-phase behavior corresponding to either the Wishart or the Gaussian noise term being dominant, depending on the value of $\beta$. Interestingly, we get a different speed of convergence for each of these phases as we show in the next theorem (the proof is given in Appendix D.1).

**Theorem 1.** *There exists a constant $C$ such that with probability at least $1 - 4e^{-n}$, if $m/n \geq C$,*

$$|\langle \hat{\boldsymbol{u}}, \boldsymbol{u} \rangle| \geq \begin{cases} 1 - \dfrac{Cnm}{(\beta/\sigma)^4}, & \text{if } \sqrt{m} > \dfrac{\beta}{\sigma} \geq (Cnm)^{\frac{1}{4}}, \\ 1 - \dfrac{Cn}{(\beta/\sigma)^2}, & \text{if } \dfrac{\beta}{\sigma} \geq \sqrt{m}, \end{cases}$$

*otherwise, $|\langle \hat{\boldsymbol{u}}, \boldsymbol{u} \rangle| \geq 1 - \frac{Cn}{(\beta/\sigma)^2}$ if $\beta/\sigma \geq \sqrt{Cn}$.*

In other words, if $\tilde{X}$ has sufficiently many more columns than rows, as the signal to noise ratio $\beta/\sigma$ increases, $\hat{\boldsymbol{u}}$ first converges to $\boldsymbol{u}$ as $1/\beta^4$, and then as $1/\beta^2$. Figure 1(a) illustrates these results. We randomly generate a rank-one $100 \times 10000$ matrix perturbed by Gaussian noise, and measure the distance between $\hat{\boldsymbol{u}}$ and $\boldsymbol{u}$. The phase transition happens at $\beta/\sigma = (nm)^{1/4}$, and there are two regimes of different convergence rates as Theorem 1 predicts.

### 2.2 Tensor Unfolding

Now let's apply the above result to the tensor version of information plus noise model studied by [20]. We consider a rank one $n \times \cdots \times n$ tensor (signal) contaminated by Gaussian noise as follows:

$$\mathcal{Y} = \mathcal{X}^* + \sigma\mathcal{E} = \beta \boldsymbol{u}^{(1)} \circ \cdots \circ \boldsymbol{u}^{(K)} + \sigma\mathcal{E}, \tag{2}$$

where factors $\boldsymbol{u}^{(k)} \in \mathbb{R}^n$, $k = 1, \ldots, K$, are unit vectors, which are not necessarily identical, and the entries of $\mathcal{E} \in \mathbb{R}^{n \times \cdots \times n}$ are i.i.d samples from the normal distribution $\mathcal{N}(0,1)$. Note that this is slightly more general (and easier to analyze) than the symmetric setting studied by [20].

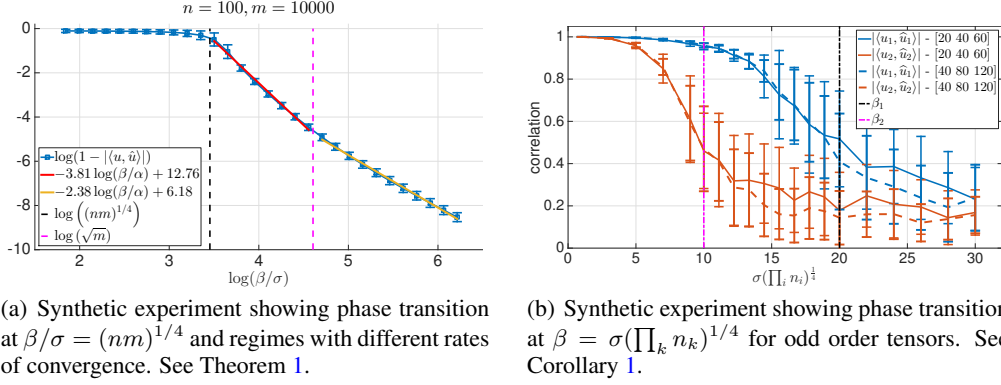

(a) Synthetic experiment showing phase transition at $\beta/\sigma = (nm)^{1/4}$ and regimes with different rates of convergence. See Theorem 1.

(b) Synthetic experiment showing phase transition at $\beta = \sigma(\prod_k n_k)^{1/4}$ for odd order tensors. See Corollary 1.

Figure 1: Numerical demonstration of Theorem 1 and Corollary 1.

Several estimators for recovering $\mathcal{X}^*$ from its noisy version $\mathcal{Y}$ have been proposed (see Table 1). Both the overlapped nuclear norm and latent nuclear norm discussed in [23] achives the relative performance guarantee

$$\|\hat{\mathcal{X}} - \mathcal{X}^*\|_F/\beta \leq O_p\left(\sigma\sqrt{n^{K-1}}/\beta\right), \tag{3}$$

where $\hat{\mathcal{X}}$ is the estimator. This bound implies that if we want to obtain relative error smaller than $\varepsilon$, we need the signal to noise ratio $\beta/\sigma$ to scale as $\beta/\sigma \succsim \sqrt{n^{K-1}}/\varepsilon$.

Mu et al. [19] proposed the square norm, defined as the nuclear norm of the matrix obtained by grouping the first $\lfloor K/2 \rfloor$ indices along the rows and the last $\lceil K/2 \rceil$ indices along the columns. This norm improves the right hand side of inequality (3) to $O_p(\sigma\sqrt{n^{\lceil K/2 \rceil}}/\beta)$, which translates to requiring $\beta/\sigma \succsim \sqrt{n^{\lceil K/2 \rceil}}/\varepsilon$ for obtaining relative error $\varepsilon$. The intuition here is the more square the unfolding is the better the bound becomes. However, there is no improvement for $K = 3$.

Richard and Montanari [20] studied the (symmetric version of) model (2) and proved that a recursive unfolding algorithm achieves the factor recovery error $\text{dist}(\hat{u}^{(k)}, u^{(k)}) = \varepsilon$ with $\beta/\sigma \succsim \sqrt{n^{\lceil K/2 \rceil}}/\varepsilon$ with high probability, where $\text{dist}(u, u') := \min(\|u - u'\|, \|u + u'\|)$. They also showed that the randomly initialized tensor power method [7, 16, 3] can achieve the same error $\varepsilon$ with slightly worse threshold $\beta/\sigma \succsim \max(\sqrt{n}/\varepsilon^2, n^{K/2})\sqrt{K \log K}$ also with high probability.

The reasoning underlying both [19] and [20] is that square unfolding is better. However, if we take the (ordinary) mode-$k$ unfolding

$$Y_{(k)} = \beta u^{(k)}\left(u^{(k-1)} \otimes \cdots \otimes u^{(1)} \otimes u^{(K)} \otimes \cdots \otimes u^{(k+1)}\right)^\top + \sigma E_{(k)}, \tag{4}$$

we can see (4) as an instance of information plus noise model (1) where $m/n = n^{K-2}$. Thus the ordinary unfolding satisfies the condition of Theorem 1 for $n$ or $K$ large enough.

**Corollary 1.** *Consider a $K(\geq 3)$th order rank one tensor contaminated by Gaussian noise as in (2). There exists a constant $C$ such that if $n^{K-2} \geq C$, with probability at least $1 - 4Ke^{-n}$, we have*

$$\text{dist}^2(\hat{u}^{(k)}, u^{(k)}) \leq \begin{cases} \dfrac{2Cn^K}{(\beta/\sigma)^4}, & \text{if } n^{\frac{K-1}{2}} > \beta/\sigma \geq C^{\frac{1}{4}}n^{\frac{K}{4}}, \\[2mm] \dfrac{2Cn}{(\beta/\sigma)^2}, & \text{if } \beta/\sigma \geq n^{\frac{K-1}{2}}, \end{cases} \quad \text{for } k = 1, \ldots, K,$$

*where $\hat{u}^{(k)}$ is the leading left singular vector of the rectangular unfolding $Y_{(k)}$.*

This proves that as conjectured by [20], the threshold $\beta/\sigma \succsim n^{K/4}$ applies not only to the even order case but also to the odd order case. Note that Hopkins et al. [10] have shown a similar result without the sharp rate of convergence. The above corollary easily extends to more general $n_1 \times \cdots \times n_K$ tensor by replacing the conditions by $\sqrt{\prod_{\ell \neq k} n_\ell} > \beta/\sigma \geq (C\prod_{k=1}^{K} n_k)^{1/4}$ and $\beta/\sigma \geq \sqrt{\prod_{\ell \neq k} n_\ell}$. The result also holds when $\mathcal{X}^*$ has rank higher than 1; see Appendix E.

We demonstrate this result in Figure 1(b). The models behind the experiment are slightly more general ones in which $[n_1, n_2, n_3] = [20, 40, 60]$ or $[40, 80, 120]$ and the signal $\mathcal{X}^*$ is rank two with $\beta_1 = 20$ and $\beta_2 = 10$. The plot shows the inner products $\langle \boldsymbol{u}_1^{(1)}, \hat{\boldsymbol{u}}_1^{(1)} \rangle$ and $\langle \boldsymbol{u}_2^{(1)}, \hat{\boldsymbol{u}}_2^{(1)} \rangle$ as a measure of the quality of estimating the two mode-1 factors. The horizontal axis is the normalized noise standard deviation $\sigma(\prod_{k=1}^{K} n_k)^{1/4}$. We can clearly see that the inner product decays symmetrically around $\beta_1$ and $\beta_2$ as predicted by Corollary 1 for both tensors.

## 3 Subspace norm for tensors

Suppose the true tensor $\mathcal{X}^* \in \mathbb{R}^{n \times \cdots \times n}$ admits a minimum Tucker decomposition [26] of rank $(R, \ldots, R)$:

$$\mathcal{X}^* = \sum_{i_1=1}^{R} \cdots \sum_{i_K=1}^{R} \beta_{i_1 i_2 \ldots i_K} \boldsymbol{u}_{i_1}^{(1)} \circ \cdots \circ \boldsymbol{u}_{i_K}^{(K)}. \tag{5}$$

If the core tensor $\mathcal{C} = (\beta_{i_1 \ldots i_K}) \in \mathbb{R}^{R \times \cdots \times R}$ is *superdiagonal*, the above decomposition reduces to the canonical polyadic (CP) decomposition [9, 15]. The mode-$k$ unfolding of the true tensor $\mathcal{X}^*$ can be written as follows:

$$\boldsymbol{X}_{(k)}^* = \boldsymbol{U}^{(k)} \boldsymbol{C}_{(k)} \left( \boldsymbol{U}^{(1)} \otimes \cdots \otimes \boldsymbol{U}^{(k-1)} \otimes \boldsymbol{U}^{(k+1)} \otimes \cdots \otimes \boldsymbol{U}^{(K)} \right)^{\top}, \tag{6}$$

where $\boldsymbol{C}_{(k)}$ is the mode-$k$ unfolding of the core tensor $\mathcal{C}$; $\boldsymbol{U}^{(k)}$ is a $n \times R$ matrix $\boldsymbol{U}^{(k)} = [\boldsymbol{u}_1^{(k)}, \ldots, \boldsymbol{u}_R^{(k)}]$ for $k = 1, \ldots, K$. Note that $\boldsymbol{U}^{(k)}$ is not necessarily orthogonal.

Let $\boldsymbol{X}_{(k)}^* = \boldsymbol{P}^{(k)} \boldsymbol{\Lambda}^{(k)} \boldsymbol{Q}^{(k)^{\top}}$ be the SVD of $\boldsymbol{X}_{(k)}^*$. We will observe that

$$\boldsymbol{Q}^{(k)} \in \text{Span}\left( \boldsymbol{P}^{(1)} \otimes \cdots \otimes \boldsymbol{P}^{(k-1)} \otimes \boldsymbol{P}^{(k+1)} \otimes \cdots \otimes \boldsymbol{P}^{(K)} \right) \tag{7}$$

because of (6) and $\boldsymbol{U}^{(k)} \in \text{Span}(\boldsymbol{P}^{(k)})$.

Corollary 1 shows that the left singular vectors $\boldsymbol{P}^{(k)}$ can be recovered under mild conditions; thus the span of the right singular vectors can also be recovered. Inspired by this, we define a norm that models a tensor $\mathcal{X}$ as a mixture of tensors $\mathcal{Z}^{(1)}, \ldots, \mathcal{Z}^{(K)}$. We require that the mode-$k$ unfolding of $\mathcal{Z}^{(k)}$, i.e. $\boldsymbol{Z}_{(k)}^{(k)}$, has a low rank factorization $\boldsymbol{Z}_{(k)}^{(k)} = \boldsymbol{M}^{(k)} \boldsymbol{S}^{(k)^{\top}}$, where $\boldsymbol{M}^{(k)} \in \mathbb{R}^{n \times H^{K-1}}$ is a variable, and $\boldsymbol{S}^{(k)} \in \mathbb{R}^{n^{K-1} \times H^{K-1}}$ is a fixed arbitrary orthonormal basis of some subspace, which we choose later to have the Kronecker structure in (7).

In the following, we define the subspace norm, suggest an approach to construct the right factor $\boldsymbol{S}^{(k)}$, and prove the denoising bound in the end.

### 3.1 The subspace norm

Consider a $K$th order tensor of size $n \times \cdots n$.

**Definition 1.** *Let* $\boldsymbol{S}^{(1)}, \ldots, \boldsymbol{S}^{(K)}$ *be matrices such that* $\boldsymbol{S}^{(k)} \in \mathbb{R}^{n^{K-1} \times H^{K-1}}$ *with* $H \leq n$. *The subspace norm for a $K$th order tensor $\mathcal{X}$ associated with* $\{\boldsymbol{S}^{(k)}\}_{k=1}^{K}$ *is defined as*

$$\|\mathcal{X}\|_s := \begin{cases} \inf_{\{\boldsymbol{M}^{(k)}\}_{k=1}^{K}} \sum_{k=1}^{K} \|\boldsymbol{M}^{(k)}\|_*, & \text{if } \mathcal{X} \in Span(\{\boldsymbol{S}^{(k)}\}_{k=1}^{K}), \\ +\infty, & \text{otherwise}, \end{cases}$$

*where* $\| \cdot \|_*$ *is the nuclear norm, and* $Span(\{\boldsymbol{S}^{(k)}\}_{k=1}^{K}) := \{\mathcal{X} \in \mathbb{R}^{n \times \cdots \times n} : \exists \boldsymbol{M}^{(1)}, \ldots, \boldsymbol{M}^{(K)}, \mathcal{X} = \sum_{k=1}^{K} \text{fold}_k(\boldsymbol{M}^{(k)} \boldsymbol{S}^{(k)^{\top}})\}$.

In the next lemma (proven in Appendix D.2), we show the dual norm of the subspace norm has a simple appealing form. As we see in Theorem 2, it avoids the $O(\sqrt{n^{K-1}})$ scaling (see the first column of Table 1) by restricting the influence of the noise term in the subspace defined by $\boldsymbol{S}^{(1)}, \ldots, \boldsymbol{S}^{(K)}$.

**Lemma 1.** *The dual norm of* $\|\cdot\|_s$ *is a semi-norm*

$$\|\mathcal{X}\|_{s^*} = \max_{k=1,\ldots,K} \|\boldsymbol{X}_{(k)} \boldsymbol{S}^{(k)}\|,$$

*where* $\| \cdot \|$ *is the spectral norm.*

## 3.2  Choosing the subspace

A natural question that arises is how to choose the matrices $\boldsymbol{S}^{(1)}, \ldots, \boldsymbol{S}^{(k)}$.

**Lemma 2.** *Let the $\boldsymbol{X}^{*}_{(k)} = \boldsymbol{P}^{(k)} \boldsymbol{\Lambda}^{(k)} \boldsymbol{Q}^{(k)}$ be the SVD of $\boldsymbol{X}^{*}_{(k)}$, where $\boldsymbol{P}^{(k)}$ is $n \times R$ and $\boldsymbol{Q}^{(k)}$ is $n^{K-1} \times R$. Assume that $R \le n$ and $\boldsymbol{U}^{(k)}$ has full column rank. It holds that for all $k$,*

  *i)* $\boldsymbol{U}^{(k)} \in \mathit{Span}(\boldsymbol{P}^{(k)})$,

  *ii)* $\boldsymbol{Q}^{(k)} \in \mathit{Span}\left( \boldsymbol{P}^{(1)} \otimes \cdots \otimes \boldsymbol{P}^{(k-1)} \otimes \boldsymbol{P}^{(k+1)} \otimes \cdots \otimes \boldsymbol{P}^{(K)} \right)$.

*Proof.* We prove the lemma in Appendix D.4. $\qquad\square$

Corollary 1 shows that when the signal to noise ratio is high enough, we can recover $\boldsymbol{P}^{(k)}$ with high probability. Hence we suggest the following three-step approach for tensor denoising:

  (i) For each $k$, unfold the observation tensor in mode $k$ and compute the top $H$ left singular vectors. Concatenate these vectors to obtain a $n \times H$ matrix $\widehat{\boldsymbol{P}}^{(k)}$.

  (ii) Construct $\boldsymbol{S}^{(k)}$ as $\boldsymbol{S}^{(k)} = \widehat{\boldsymbol{P}}^{(1)} \otimes \cdots \otimes \widehat{\boldsymbol{P}}^{(k-1)} \otimes \widehat{\boldsymbol{P}}^{(k+1)} \otimes \cdots \otimes \widehat{\boldsymbol{P}}^{(K)}$.

  (iii) Solve the subspace norm regularized minimization problem

$$\min_{\mathcal{X}} \quad \frac{1}{2}\|\mathcal{Y} - \mathcal{X}\|_F^2 + \lambda \|\mathcal{X}\|_s, \tag{8}$$

   where the subspace norm is associated with the above defined $\{\boldsymbol{S}^{(k)}\}_{k=1}^{K}$.

See Appendix B for details.

## 3.3  Analysis

Let $\mathcal{Y} \in \mathbb{R}^{n \times \cdots \times n}$ be a tensor corrupted by Gaussian noise with standard deviation $\sigma$ as follows:

$$\mathcal{Y} = \mathcal{X}^* + \sigma \mathcal{E}. \tag{9}$$

We define a slightly modified estimator $\hat{\mathcal{X}}$ as follows:

$$\hat{\mathcal{X}} = \operatorname*{arg\,min}_{\mathcal{X}, \{\boldsymbol{M}^{(k)}\}_{k=1}^{K}} \left\{ \frac{1}{2}\|\mathcal{Y} - \mathcal{X}\|_F^2 + \lambda \|\mathcal{X}\|_s : \ \mathcal{X} = \sum_{k=1}^{K} \operatorname{fold}_k \left( \boldsymbol{M}^{(k)} \boldsymbol{S}^{(k)\top} \right), \{\boldsymbol{M}^{(k)}\}_{k=1}^{K} \in \mathcal{M}(\rho) \right\} \tag{10}$$

where $\mathcal{M}(\rho)$ is a restriction of the set of matrices $\boldsymbol{M}^{(k)} \in \mathbb{R}^{n \times H^{K-1}}$, $k = 1, \ldots, K$ defined as follows:

$$\mathcal{M}(\rho) := \left\{ \{\boldsymbol{M}^{(k)}\}_{k=1}^{K} : \|\operatorname{fold}_k(\boldsymbol{M}^{(k)})_{(\ell)}\| \le \frac{\rho}{K}(\sqrt{n} + \sqrt{H^{K-1}}), \forall k \ne \ell \right\}.$$

This restriction makes sure that $\boldsymbol{M}^{(k)}$, $k = 1, \ldots, K$, are incoherent, i.e., each $\boldsymbol{M}^{(k)}$ has a spectral norm that is as low as a random matrix when unfolded at a different mode $\ell$. Similar assumptions were used in low-rank plus sparse matrix decomposition [2, 12] and for the denoising bound for the latent nuclear norm [23].

Then we have the following statement (we prove this in Appendix D.3).

**Theorem 2.** *Let $\mathcal{X}_p$ be any tensor that can be expressed as*

$$\mathcal{X}_p = \sum_{k=1}^{K} \operatorname{fold}_k \left( \boldsymbol{M}_p^{(k)} \boldsymbol{S}^{(k)\top} \right),$$

*which satisfies the above incoherence condition $\{\boldsymbol{M}_p^{(k)}\}_{k=1}^{K} \in \mathcal{M}(\rho)$ and let $r_k$ be the rank of $\boldsymbol{M}_p^{(k)}$ for $k = 1, \ldots, K$. In addition, we assume that each $\boldsymbol{S}^{(k)}$ is constructed as $\boldsymbol{S}^{(k)} =$*

$\widehat{\boldsymbol{P}}^{(k-1)} \otimes \cdots \otimes \widehat{\boldsymbol{P}}^{(k+1)}$ *with* $(\widehat{\boldsymbol{P}}^{(k)})^\top \widehat{\boldsymbol{P}}^{(k)} = \boldsymbol{I}_H$. *Then there are universal constants* $c_0$ *and* $c_1$ *such that any solution* $\hat{\mathcal{X}}$ *of the minimization problem* (10) *with* $\lambda = \|\mathcal{X}_p - \mathcal{X}^*\|_{s^*} + c_0 \sigma \left( \sqrt{n} + \sqrt{H^{K-1}} + \sqrt{2\log(K/\delta)} \right)$ *satisfies the following bound*

$$\|\hat{\mathcal{X}} - \mathcal{X}^*\|_F \leq \|\mathcal{X}_p - \mathcal{X}^*\|_F + c_1 \lambda \sqrt{\sum\nolimits_{k=1}^{K} r_k},$$

*with probability at least* $1 - \delta$.

Note that the right-hand side of the bound consists of two terms. The first term is the approximation error. This term will be zero if $\mathcal{X}^*$ lies in $\mathrm{Span}(\{\boldsymbol{S}^{(k)}\}_{k=1}^K)$. This is the case, if we choose $\boldsymbol{S}^{(k)} = \boldsymbol{I}_{n^{K-1}}$ as in the latent nuclear norm, or if the condition of Corollary 1 is satisfied for the smallest $\beta_R$ when we use the Kronecker product construction we proposed. Note that the regularization constant $\lambda$ should also scale with the dual subspace norm of the residual $\mathcal{X}_p - \mathcal{X}^*$.

The second term is the estimation error with respect to $\mathcal{X}_p$. If we take $\mathcal{X}_p$ to be the orthogonal projection of $\mathcal{X}^*$ to the $\mathrm{Span}(\{\boldsymbol{S}^{(k)}\}_{k=1}^K)$, we can ignore the contribution of the residual to $\lambda$, because $(\mathcal{X}_p - \mathcal{X}^*)_{(k)} \boldsymbol{S}^{(k)} = 0$. Then the estimation error scales mildly with the dimensions $n$, $H^{K-1}$ and with the sum of the ranks. Note that if we take $\boldsymbol{S}^{(k)} = \boldsymbol{I}_{n^{K-1}}$, we have $H^{K-1} = n^{K-1}$, and we recover the guarantee (3) .

## 4 Experiments

In this section, we conduct tensor denoising experiments on synthetic and real datasets, to numerically confirm our analysis in previous sections.

### 4.1 Synthetic data

We randomly generated the true rank two tensor $\mathcal{X}^*$ of size $20 \times 30 \times 40$ with singular values $\beta_1 = 20$ and $\beta_2 = 10$. The true factors are generated as random matrices with orthonormal columns. The observation tensor $\mathcal{Y}$ is then generated by adding Gaussian noise with standard deviation $\sigma$ to $\mathcal{X}^*$.

Our approach is compared to the CP decomposition, the overlapped approach, and the latent approach. The CP decomposition is computed by the tensorlab [22] with 20 random initializations. We assume CP knows the true rank is 2. For the subspace norm, we use Algorithm 2 described in Section 3. We also select the top 2 singular vectors when constructing $\widehat{\boldsymbol{U}}^{(k)}$'s. We computed the solutions for 20 values of regularization parameter $\lambda$ logarithmically spaced between 1 and 100. For the overlapped and the latent norm, we use ADMM described in [25]; we also computed 20 solutions with the same $\lambda$'s used for the subspace norm.

We measure the performance in the relative error defined as $\|\widehat{\mathcal{X}} - \mathcal{X}^*\|_F / \|\mathcal{X}^*\|_F$. We report the minimum error obtained by choosing the optimal regularization parameter or the optimal initialization. Although the regularization parameter could be selected by leaving out some entries and measuring the error on these entries, we will not go into tensor completion here for the sake of simplicity.

Figure 2 (a) and (b) show the result of this experiment. The left panel shows the relative error for 3 representative values of $\lambda$ for the subspace norm. The black dash-dotted line shows the minimum error across all the $\lambda$'s. The magenta dashed line shows the error corresponding to the theoretically motivated choice $\lambda = \sigma(\max_k(\sqrt{n_k} + \sqrt{H^{K-1}}) + \sqrt{2\log(K)})$ for each $\sigma$. The two vertical lines are thresholds of $\sigma$ from Corollary 1 corresponding to $\beta_1$ and $\beta_2$, namely, $\beta_1/(\prod_k n_k)^{1/4}$ and $\beta_2/(\prod_k n_k)^{1/4}$. It confirms that there is a rather sharp increase in the error around the theoretically predicted places (see also Figure 1(b)). We can also see that the optimal $\lambda$ should grow linearly with $\sigma$. For large $\sigma$ (small SNR), the best relative error is 1 since the optimal choice of the regularization parameter $\lambda$ leads to predicting with $\widehat{\mathcal{X}} = 0$.

Figure 2 (b) compares the performance of the subspace norm to other approaches. For each method the smallest error corresponding to the optimal choice of the regularization parameter $\lambda$ is shown.

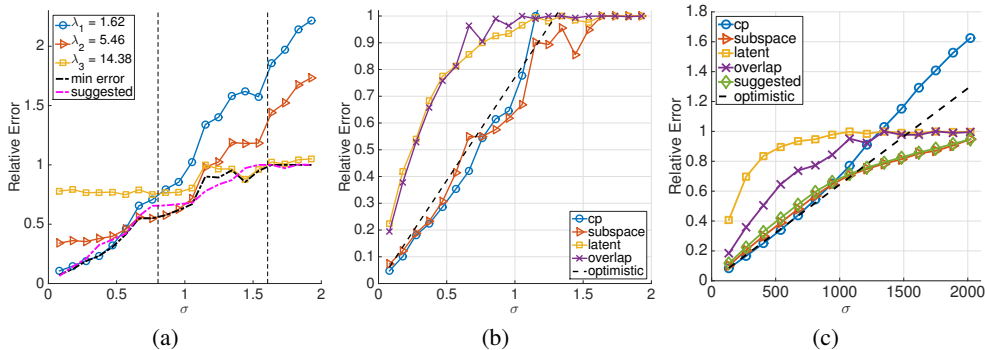

Figure 2: Tensor denoising. (a) The subspace approach with three representative $\lambda$'s on synthetic data. (b) Comparison of different methods on synthetic data. (c) Comparison on amino acids data.

In addition, to place the numbers in context, we plot the line corresponding to

$$\text{Relative error} = \frac{\sqrt{R \sum_k n_k \log(K)}}{\|\mathcal{X}^*\|_F} \cdot \sigma, \tag{11}$$

which we call "optimistic". This can be motivated from considering the (non-tractable) maximum likelihood estimator for CP decomposition (see Appendix A).

Clearly, the error of CP, the subspace norm, and "optimistic" grows at the same rate, much slower than overlap and latent. The error of CP increases beyond 1, as no regularization is imposed (see Appendix C for more experiments). We can see that both CP and the subspace norm are behaving near optimally in this setting, although such behavior is guaranteed for the subspace norm whereas it is hard to give any such guarantee for the CP decomposition based on nonlinear optimization.

## 4.2 Amino acids data

The amino acid dataset [5] is a semi-realistic dataset commonly used as a benchmark for low rank tensor modeling. It consists of five laboratory-made samples, each one contains different amounts of tyrosine, tryptophan and phenylalanine. The spectrum of their excitation wavelength (250-300 nm) and emission (250-450 nm) are measured by fluorescence, which gives a $5 \times 201 \times 61$ tensor. As the true factors are known to be these three acids, this data perfectly suits the CP model. The true rank is fed into CP and the proposed approach as $H = 3$. We computed the solutions of CP for 20 different random initializations, and the solutions of other approaches with 20 different values of $\lambda$. For the subspace and the overlapped approach, $\lambda$'s are logarithmically spaced between $10^3$ and $10^5$. For the latent approach, $\lambda$'s are logarithmically spaced between $10^4$ and $10^6$. Again, we include the optimistic scaling (11) to put the numbers in context.

Figure 2(c) shows the smallest relative error achieved by all methods we compare. Similar to the synthetic data, both CP and the subspace norm behaves near ideally, though the relative error of CP can be larger than 1 due to the lack of regularization. Interestingly the theoretically suggested scaling of the regularization parameter $\lambda$ is almost optimal.

## 5 Conclusion

We have settled a conjecture posed by [20] and showed that indeed $O(n^{K/4})$ signal-to-noise ratio is sufficient also for odd order tensors. Moreover, our analysis shows an interesting two-phase behavior of the error. This finding lead us to the development of the proposed subspace norm. The proposed norm is defined with respect to a set of orthonormal matrices $\widehat{\boldsymbol{P}}^{(1)}, \ldots, \widehat{\boldsymbol{P}}^{(K)}$, which are estimated by mode-wise singular value decompositions. We have analyzed the denoising performance of the proposed norm, and shown that the error can be bounded by the sum of two terms, which can be interpreted as an approximation error term coming from the first (non-convex) step, and an estimation error term coming from the second (convex) step.

## Footnotes

[1] We say $a_n \gtrsim b_n$ if there is a constant $C > 0$ such that $a_n \geq C \cdot b_n$.
