[Supplementary Material]

# A Maximum likelihood estimator

Let $\mathcal{Y} \in \mathbb{R}^{n_1 \times \cdots \times n_K}$ be a noisy observed tensor generated as follows:

$$\mathcal{Y} = \mathcal{X}^* + \sigma\mathcal{E} = \sum_{r=1}^{R} \beta_r \boldsymbol{u}_r^{(1)} \circ \cdots \circ \boldsymbol{u}_r^{(K)} + \sigma\mathcal{E},$$

where $\mathcal{E}$ is a noisy tensor whose entries are i.i.d. normal $\mathcal{N}(0,1)$.

Let $\hat{\mathcal{X}}_{\text{MLE}}$ be the (intractable) estimator defined as

$$\hat{\mathcal{X}}_{\text{MLE}} = \arg\min_{\mathcal{X}} \left( \|\mathcal{Y} - \mathcal{X}\|_F^2 : \text{rank}(\mathcal{X}) \leq R \right).$$

We have the following performance guarantee for $\hat{\mathcal{X}}_{\text{MLE}}$:

**Theorem 3.** *Let $R \leq \min_k n_k/2$. Then there is a constant $c$ such that*

$$\|\hat{\mathcal{X}}_{MLE} - \mathcal{X}^*\|_F \leq c\sigma\sqrt{R^K \sum_{k=1}^{K} n_k \log(2K/K_0) + \log(2/\delta)},$$

*with probability at least $1 - \delta$, where $K_0 = \log(3/2)$.*

Note that the factor $R^K$ in the square root is rather conservative. In the best case, this factor reduces to linear in $R$ and this is what we present in Section 4 as "optimistic" ignoring constants and $\delta$; see Eq. (11).

*Proof of Theorem 3.* Since $\hat{\mathcal{X}}_{\text{MLE}}$ is a minimizer and $\mathcal{X}^*$ is also feasible, we have

$$\|\mathcal{Y} - \hat{\mathcal{X}}_{\text{MLE}}\|_F^2 \leq \|\mathcal{Y} - \mathcal{X}^*\|_F^2,$$

which implies

$$\|\mathcal{X}^* - \hat{\mathcal{X}}_{\text{MLE}}\|_F^2 \leq \sigma\langle\mathcal{E}, \hat{\mathcal{X}}_{\text{MLE}} - \mathcal{X}^*\rangle$$
$$\leq \sigma\|\mathcal{E}\|_{\text{op}}\|\hat{\mathcal{X}}_{\text{MLE}} - \mathcal{X}^*\|_{\text{nuc}},$$

where

$$\|\mathcal{X}\|_{\text{op}} := \sup_{\boldsymbol{u}^{(1)},\ldots,\boldsymbol{u}^{(K)}} \Big\{ \sum_{i_1,i_2,\ldots,i_K} \mathcal{X}_{i_1,i_2,\ldots,i_K} u_{i_1}^{(1)} u_{i_2}^{(2)} \cdots u_{i_K}^{(K)} :$$
$$\|\boldsymbol{u}^{(1)}\| = \|\boldsymbol{u}^{(2)}\| = \cdots = \|\boldsymbol{u}^{(K)}\| = 1 \Big\}$$

is the tensor spectral norm and the nuclear norm

$$\|\mathcal{X}\|_{\text{nuc}} := \inf_{\boldsymbol{u}^{(1)},\ldots,\boldsymbol{u}^{(K)}} \Big\{ \sum_r \|\boldsymbol{u}_r^{(1)}\| \cdot \|\boldsymbol{u}_r^{(2)}\| \cdots \|\boldsymbol{u}_r^{(K)}\| :$$
$$\mathcal{X} = \sum_{r=1}^{R} \boldsymbol{u}_r^{(1)} \circ \cdots \circ \boldsymbol{u}_r^{(K)} \Big\}$$

is the dual of the spectral norm.

Since both $\hat{\mathcal{X}}_{\text{MLE}}$ and $\mathcal{X}^*$ are rank at most $R$, the difference $\hat{\mathcal{X}}_{\text{MLE}} - \mathcal{X}^*$ is rank at most $2R$. Moreover, any rank-$R$ CP decomposition with $R \leq \min_k n_k$ can be reduced to an orthogonal CP decomposition with rank at most $R^K$ via the Tucker decomposition [14]. Thus, denoting this orthogonal decomposition by $\hat{\mathcal{X}}_{\text{MLE}} - \mathcal{X}^* = \sum_{r=1}^{R^K} \tilde{\boldsymbol{u}}_r^{(1)} \circ \cdots \circ \tilde{\boldsymbol{u}}_r^{(K)}$ and using $\beta_r := \|\tilde{\boldsymbol{u}}_r^{(1)}\| \cdots \|\tilde{\boldsymbol{u}}_r^{(K)}\|$, we have

$$\|\hat{\mathcal{X}}_{\text{MLE}} - \mathcal{X}^*\|_{\text{nuc}} \leq \sum_{r=1}^{R^K} \beta_r \leq \sqrt{R^K}\sqrt{\sum_{r=1}^{R^K} \beta_r^2}$$
$$= \sqrt{R^K}\|\hat{\mathcal{X}}_{\text{MLE}} - \mathcal{X}^*\|_F,$$

where the last equality follows because the decomposition is orthogonal.

Finally applying the tail bound for the spectral norm $\|\mathcal{E}\|_{\text{op}}$ of random Gaussian tensor $\mathcal{E}$ [24], we obtain what we wanted. $\square$

---
**Algorithm 1:** Tensor denoising via the subspace norm
___
   **Input:** noisy tensor $\mathcal{Y}$, subspace dimension $H$, regularization constant $\lambda$

   **for** $k = 1$ **to** $K$ **do**

      $\widehat{\boldsymbol{P}}^{(k)} \longleftarrow$ top $H$ left singular vectors of $\boldsymbol{Y}_{(k)}$

   **end for**

   **for** $k = 1$ **to** $K$ **do**

      $\boldsymbol{S}^{(k)} \longleftarrow \widehat{\boldsymbol{P}}^{(1)} \otimes \cdots \otimes \widehat{\boldsymbol{P}}^{(k-1)} \otimes \widehat{\boldsymbol{P}}^{(k+1)} \otimes \cdots \otimes \widehat{\boldsymbol{P}}^{(K)}$

   **end for**

   **Output:** $\widehat{\mathcal{X}} = \arg\min_{\mathcal{X}} \frac{1}{2}\|\mathcal{Y} - \mathcal{X}\|_F^2 + \lambda\|\mathcal{X}\|_s$.
___

## B   Details of optimization

For solving problem (8), we follow the alternating direction method of multipliers described in [25]. We scale the objective function in (8) by $1/\lambda$, and consider the dual problem

$$\min_{\mathcal{D},\{\boldsymbol{W}^{(k)}\}_{k=1}^K} \quad \frac{\lambda}{2}\|\mathcal{D}\|_F^2 - \langle \mathcal{D}, \mathcal{Y} \rangle$$

$$\text{s.t.} \quad \max_k \|\boldsymbol{W}^{(k)}\| \leq 1, \tag{12}$$

$$\boldsymbol{W}^{(k)} = \boldsymbol{D}_{(k)}\boldsymbol{S}^{(k)}, \ k = 1, \ldots, K,$$

where $\mathcal{D} \in \mathbb{R}^{n_1 \times n_2 \times \cdots \times n_K}$ is the dual tensor that corresponds to the residual in the primal problem (8), and $\boldsymbol{W}^{(k)}$'s are auxiliary variables introduced to make the problem equality constrained.

The augmented Lagrangian function of problem (12) could be written as follows:

$$L_\eta(\mathcal{D}, \{\boldsymbol{W}^{(k)}\}_{k=1}^K, \{\boldsymbol{M}^{(k)}\}_{k=1}^K)$$

$$= \frac{\lambda}{2}\|\mathcal{D}\|^2 - \langle \mathcal{D}, \mathcal{Y} \rangle + \sum_{k=1}^K \left( \langle \boldsymbol{M}^{(k)}, \boldsymbol{D}_{(k)}\boldsymbol{S}^{(k)} - \boldsymbol{W}^{(k)} \rangle \right.$$

$$\left. + \frac{\eta}{2}\|\boldsymbol{D}_{(k)}\boldsymbol{S}^{(k)} - \boldsymbol{W}^{(k)}\|_F^2 + \mathbf{1}_{\|\cdot\|\leq 1}(\boldsymbol{W}^{(k)}) \right),$$

where $\boldsymbol{M}^{(k)}$'s are the multipliers, $\eta$ is the augmenting parameter, and $\mathbf{1}_{\|\cdot\|\leq 1}$ is the indicator function of the unit spectral norm ball.

We follow the derivation in [25] and conclude that the updates of $\mathcal{D}$, $\boldsymbol{M}^{(k)}$ and $\boldsymbol{W}^{(k)}$ can be computed in closed forms. We further combine the updates of $\boldsymbol{W}^{(k)}$ and other steps so that it needs not to be explicitly computed. The sum of the products of $\boldsymbol{M}^{(k)}$ and $\boldsymbol{S}^{(k)^\top}$ finally converges to the solution of the primal problem (8), see Algorithm 2.

The update for the Lagrangian multipliers $\boldsymbol{M}^{(k)}$ ($k = 1, \ldots, K$) is written as singular value soft-thresholding operator defined as

$$\text{prox}_\eta^{tr}(\boldsymbol{Z}) = \boldsymbol{P}\max(\boldsymbol{\Sigma} - \eta, 0)\boldsymbol{Q}^\top,$$

where $\boldsymbol{Z} = \boldsymbol{P}\boldsymbol{\Sigma}\boldsymbol{Q}^\top$ is the SVD of $\boldsymbol{Z}$.

A notable property of the subspace norm is the computational efficiency. The update of $\boldsymbol{M}^{(k)}$ requires singular value decomposition, which usually dominates the costs of computation. For problem (12), the size of $\boldsymbol{M}^{(k)}$ is only $n_k \times H^{K-1}$. Comparing with previous approaches, e.g. the latent approach whose multipliers are $n_k \times \prod_{k' \neq k} n_{k'}$ matrices, the size of our variables is much smaller, so the per-iteration cost is reduced.

## C   Additional experiments

We report the experimental results when the input rank of CP and the subspace approach is are over-specified, on the same synthetic dataset as Section 4. We consider the case where the input rank is 8.

---

**Algorithm 2:** ADMM for subspace norm minimization

---

**Input:** $\mathcal{Y}$, $\lambda$, $\boldsymbol{S}^{(1)}, \ldots, \boldsymbol{S}^{(K)}$, $\eta$, initializations $\mathcal{D}_0$, $\{\boldsymbol{M}_0^{(1)}, \ldots, \boldsymbol{M}_0^{(K)}\}$

$t = 0$

**repeat**

$$\mathcal{D}_{t+1} = \frac{1}{\lambda + \eta K}\left(\mathcal{Y} + K\eta\mathcal{D}_t - \sum_k \mathrm{fold}_k\big((2\boldsymbol{M}_t^{(k)} - \boldsymbol{M}_{t-1}^{(k)})\boldsymbol{S}^{(k)\top}\big)\right)$$

    **for** $k = 1$ **to** $K$ **do**

$$\boldsymbol{M}_{t+1}^{(k)} = \mathrm{prox}_\eta^{tr}\left(\boldsymbol{M}_t^{(k)} + \eta\boldsymbol{D}_{(k),t+1}\boldsymbol{S}^{(k)}\right)$$

    **end for**

    $t \leftarrow t + 1$

**until** convergence

**Output:** $\widehat{\mathcal{X}} = \sum_{k=1}^K \boldsymbol{M}_t^{(k)}\boldsymbol{S}^{(k)\top}$.

---

Figure 3: Tensor denoising on synthetic dataset when the input rank is larger than the truth.

We impose the $\ell_2$ regularizations on the factors of CP. We test 20 values that are logarithmically spaced between 0.01 and 10 are the regularization parameter. For each value, we compute 20 solutions with random initializations and select the one with lowest objective value.

For the subspace approach, we computed solutions for 20 values of the regularization parameter that are logarithmically spaced between 1 and 1000.

As before, we report the minimum relative error obtained by the same method. The results are shown in Figure 3. We include the case the rank is specified incorrectly for comparison. Clearly, even if the rank is much larger than the truth, the subspace approach and CP are robust with proper regularization.

# D Proofs

## D.1 Proof of Theorem 1

We consider the second moment of $\tilde{X}$:

$$\tilde{X}\tilde{X}^\top = \beta^2 uu^\top + \sigma^2 EE^\top + \beta\sigma(uv^\top E^\top + Evu^\top)$$

$$= \overbrace{\beta^2 uu^\top + m\sigma^2 I}^{B} +$$

$$\overbrace{\sigma^2 EE^\top - m\sigma^2 I + \beta\sigma(uv^\top E^\top + Evu^\top)}^{G}.$$

The eigenvalue decomposition of $B$ can be written as

$$B = [u\ U_2] \begin{bmatrix} \beta^2 + m\sigma^2 & \\ & m\sigma^2 I \end{bmatrix} \begin{bmatrix} u^\top \\ U_2^\top \end{bmatrix}.$$

We first show a deterministic lower bound for $|\langle \hat{u}, u\rangle|$ assuming $\beta^2 \geq 2\|G\|$, where $\hat{u}$ is the leading eigenvector of $\tilde{X}\tilde{X}^\top$. Then we bound the spectral norm $\|G\|$ of the noise term (Lemma 3) and derive the sufficient condition for $\beta$.

Let $\hat{u}$ be the leading eigenvector of $\tilde{X}\tilde{X}^\top$ with eigenvalue $\hat{\lambda}$, $r = B\hat{u} - \hat{\lambda}\hat{u} = -G\hat{u}$. We have $U_2^\top r = (m\sigma^2 - \hat{\lambda})U_2^\top \hat{u}$. Hence, for all $\beta^2 > 2\|G\|$, it holds that

$$|\sin(\hat{u}, u)| = \|U_2^\top \hat{u}\|_2 = \frac{\|U_2^\top r\|_2}{\hat{\lambda} - m\sigma^2} \leq \frac{\|G\|}{\beta^2 - \|G\|} \leq \frac{2\|G\|}{\beta^2},$$

where we used $\|U_2^\top r\|_2 = \|U_2^\top G\hat{u}\|_2 \leq \|G\|$, and $\hat{\lambda} \geq u^\top \tilde{X}\tilde{X}^\top u^\top \geq \beta^2 + m\sigma^2 - \|G\|$. Therefore,

$$|\langle \hat{u}, u\rangle| = |\cos(\hat{u}, u)| \geq \sqrt{1 - \frac{4\|G\|^2}{\beta^4}} \geq 1 - \frac{4\|G\|^2}{\beta^4},$$

if $\beta^2 \geq 2\|G\|$.

It follows from Lemma 3 (shown below) that

$$\|G\| \leq \begin{cases} 2\bar{C}\sigma^2\sqrt{mn}, & \text{if } \beta/\sigma < \sqrt{m}, \\ 2\bar{C}\beta\sigma\sqrt{n}, & \text{otherwise}, \end{cases}$$

where $\bar{C}$ is a universal constant with probability at least $1 - 4e^{-n}$.

Now consider the first case ($\beta/\sigma < \sqrt{m}$) and assume $\beta^2 \geq 4\bar{C}\sigma^2\sqrt{mn} \geq 2\|G\|$. Note that this case only arises when $\sqrt{m} \geq 4\bar{C}\sqrt{n}$. Denoting $C = 16\bar{C}^2$, we obtain the first case in the theorem. Next, consider the second case ($\beta/\sigma \geq \sqrt{m}$). If $\sqrt{m} \geq 4\bar{C}\sqrt{n}$ as above, we have $\beta/\sigma \geq 4\bar{C}\sqrt{n}$, which implies $\beta^2 \geq 2\|G\|$ and we obtain the second case in the theorem. On the other hand, if $\sqrt{m} < 4\bar{C}\sqrt{n}$, we require $\beta/\sigma \geq 4\bar{C}\sqrt{n}$ to obtain the last case in the theorem.

**Lemma 3.** *Let $G$ be constructed as in Theorem 1. If $m \geq n$, there exists an universal constant $\bar{C}$ such that*

$$\|G\| \leq \bar{C}\sigma^2\left(\sqrt{mn} + \sqrt{n(\beta/\sigma)^2}\right),$$

*with probability at least $1 - 4e^{-n}$.*

*Proof.* The proof is an $\varepsilon$-net argument. Let

$$\lambda = 2\sigma^2\left(\sqrt{4mn} + 4n + \sqrt{8n(\beta/\sigma)^2}\right).$$

The goal is to control $|x^\top Gx|$ for all the vectors $x$ on the unit Euclidean sphere $\mathcal{S}^{n-1}$. In order to do this, we first bound the probability of the tail event $|x^\top Gx| > \lambda$, for any fixed $x \in \mathcal{S}^{n-1}$. Then

we bound the probability that $|\boldsymbol{x}^\top \boldsymbol{G}\boldsymbol{x}| > \lambda$ for all the vectors in a $\varepsilon$-net $\mathcal{N}_\varepsilon$. Finally, we establish the connection between $\sup_{\boldsymbol{x} \in \mathcal{N}_\varepsilon} |\boldsymbol{x}^\top \boldsymbol{G}\boldsymbol{x}|$ and $\|\boldsymbol{G}\|$.

To bound $\mathbb{P}(|\boldsymbol{x}^\top \boldsymbol{G}\boldsymbol{x}| > \lambda)$ for a fix $\boldsymbol{x} \in \mathcal{S}^{n-1}$, we expand $\boldsymbol{x}^\top \boldsymbol{G}\boldsymbol{x}$ as
$$\boldsymbol{x}^\top \boldsymbol{G}\boldsymbol{x} = \sigma^2(\|\boldsymbol{z}\|^2 - m) + 2\beta\sigma(\boldsymbol{u}^\top \boldsymbol{x})\gamma,$$
where $\boldsymbol{z} = \boldsymbol{E}^\top \boldsymbol{x}$ and $\gamma = \boldsymbol{v}^\top \boldsymbol{z}$. Since $\boldsymbol{z} \sim \mathcal{N}(0, \boldsymbol{I})$, we can see that $\|\boldsymbol{z}\|^2$ is $\chi^2$ distributed with $m$ degrees of freedom and $\gamma \sim \mathcal{N}(0, 1)$.

First we bound the deviation of the $\chi^2$ term. By the corollary of Lemma 1 in [17], we have
$$\mathbb{P}(\left|\|\boldsymbol{z}\|^2 - m\right| > \lambda_1) \le 2e^{-4n}, \tag{13}$$
where $\lambda_1 = 2(\sqrt{4mn} + 4n)$.

Next we bound the deviation of the Gaussian term. Using the Gaussian tail inequality, we have
$$\mathbb{P}(|\gamma| > \lambda_2) \le 2e^{-4n}, \tag{14}$$
where $\lambda_2 = \sqrt{8n}$.

Combining inequalities (22) and (14), we have
$$\begin{aligned}
&\mathbb{P}(|\boldsymbol{x}^\top \boldsymbol{G}\boldsymbol{x}| > \lambda) \\
&\le \mathbb{P}\left(\sigma^2\left|\|\boldsymbol{z}\|^2 - m\right| + |2\beta\sigma(\boldsymbol{u}^\top \boldsymbol{x})\gamma| > \sigma^2\lambda_1 + 2\beta\sigma\lambda_2\right) \\
&\le \mathbb{P}\left(\left|\|\boldsymbol{z}\|^2 - m\right| > \lambda_1 \lor |\gamma| > \lambda_2\right) \\
&\le \mathbb{P}\left(\left|\|\boldsymbol{z}\|^2 - m\right| > \lambda_1\right) + \mathbb{P}(|\gamma| > \lambda_2) \\
&\le 4e^{-4n},
\end{aligned}$$
where the second to last line follows from the union bound.

Furthermore, using Lemma 5.2 and 5.4 of [27], for any $\varepsilon \in [0, 1)$, it holds that
$$|\mathcal{N}_\varepsilon| \le (1 + 2/\varepsilon)^n,$$
and
$$\|\boldsymbol{G}\| \le (1 - 2\varepsilon)^{-1} \sup_{\boldsymbol{x} \in \mathcal{N}_\varepsilon} |\boldsymbol{x}^\top \boldsymbol{G}\boldsymbol{x}|.$$
Taking the union bound over all the vectors in $\mathcal{N}_{1/4}$, we obtain
$$\mathbb{P}\left(\sup_{\boldsymbol{x} \in \mathcal{N}_{1/4}} |\boldsymbol{x}^\top \boldsymbol{G}\boldsymbol{x}| > \lambda\right) \le |\mathcal{N}_{1/4}| 4e^{-4n} < 4e^{-n}.$$

Finally, the statement is obtained by noticing that $n \le m$. $\qquad\square$

We prove a more general version of the theorem that allows the signal part to be rank $R$ in Appendix E.

### D.2  Proof of Lemma 1

*Proof.* By definition,

$$\begin{aligned}
\|\mathcal{Y}\|_{s*} &= \sup_{\{\boldsymbol{M}^{(k)}\}_{k=1}^K} \langle \mathcal{Y}, \sum_{k=1}^K \mathrm{fold}_k(\boldsymbol{M}^{(k)}\boldsymbol{S}^{(k)^\top})\rangle \\
&\quad \text{s.t.} \sum_{k=1}^K \|\boldsymbol{M}^{(k)}\|_* \le 1 \\
&= \sup_{\{\boldsymbol{M}^{(k)}\}_{k=1}^K} \sum_{k=1}^K \langle \boldsymbol{Y}_{(k)}\boldsymbol{S}^{(k)}, \boldsymbol{M}^{(k)}\rangle \\
&\quad \text{s.t.} \sum_{k=1}^K \|\boldsymbol{M}^{(k)}\|_* \le 1 \\
&= \max_k \|\boldsymbol{Y}_{(k)}\boldsymbol{S}^{(k)}\|,
\end{aligned}$$

where we used the Hölder inequality in the last line. □

### D.3 Proof of Theorem 2

First we decompose the error as

$$\|\mathcal{X}^* - \hat{\mathcal{X}}\|_F \leq \|\mathcal{X}^* - \mathcal{X}_p\|_F + \|\mathcal{X}_p - \hat{\mathcal{X}}\|_F.$$

The first term is an approximation error that depends on the choice of the subspace $\boldsymbol{S}^{(k)}$. The second term corresponds to an estimation error and we analyze the second term below.

Since $\hat{\mathcal{X}}$ is the minimizer of (10) and $\mathcal{X}_p$ is feasible,

$$\frac{1}{2}\|\mathcal{Y} - \hat{\mathcal{X}}\|_F^2 + \lambda \sum_{k=1}^{K} \|\hat{\boldsymbol{M}}^{(k)}\|_* \leq \frac{1}{2}\|\mathcal{Y} - \mathcal{X}_p\|_F^2 + \lambda \sum_{k=1}^{K} \|\boldsymbol{M}_p^{(k)}\|_*,$$

from which we have

$$\frac{1}{2}\|\mathcal{X}_p - \hat{\mathcal{X}}\|_F^2 \leq \|\mathcal{Y} - \mathcal{X}_p\|_{s^*}\|\mathcal{X}_p - \hat{\mathcal{X}}\|_s + \lambda \sum_{k=1}^{K} \left( \|\boldsymbol{M}_p^{(k)}\|_* - \|\hat{\boldsymbol{M}}^{(k)}\|_* \right). \qquad (15)$$

Next we define $\boldsymbol{\Delta}_k := \hat{\boldsymbol{M}}^{(k)} - \boldsymbol{M}_p^{(k)} \in \mathbb{R}^{n_k \times H^{K-1}}$ and define its orthogonal decomposition $\boldsymbol{\Delta}_k = \boldsymbol{\Delta}_k' + \boldsymbol{\Delta}_k''$ as

$$\boldsymbol{\Delta}_k'' := (\boldsymbol{I}_{n_k} - \boldsymbol{P}_{U_p})\boldsymbol{\Delta}_k(\boldsymbol{I}_{H^{K-1}} - \boldsymbol{P}_{V_p}),$$

where $\boldsymbol{P}_{U_p}$ and $\boldsymbol{P}_{V_p}$ are projection matrices to the column and row spaces of $\boldsymbol{M}_p^{(k)}$, respectively, and $\boldsymbol{\Delta}_k' := \boldsymbol{\Delta}_k - \boldsymbol{\Delta}_k''$.

The above definition allows us to decompose $\|\hat{\boldsymbol{M}}^{(k)}\|_*$ as follows:

$$\|\hat{\boldsymbol{M}}^{(k)}\|_* = \|\boldsymbol{M}_p^{(k)} + \boldsymbol{\Delta}_k'' + \boldsymbol{\Delta}_k'\|_*$$
$$\geq \|\boldsymbol{M}_p^{(k)}\|_* + \|\boldsymbol{\Delta}_k''\|_* - \|\boldsymbol{\Delta}_k'\|_*. \qquad (16)$$

Moreover,

$$\|\mathcal{X}_p - \hat{\mathcal{X}}\|_s \leq \sum_{k=1}^{K} \|\boldsymbol{\Delta}_k\|_* \leq \sum_{k=1}^{K} \left( \|\boldsymbol{\Delta}_k'\|_* + \|\boldsymbol{\Delta}_k''\|_* \right) \qquad (17)$$

Combining inequalities (15)–(17), we have

$$\frac{1}{2}\|\mathcal{X}_p - \hat{\mathcal{X}}\|_F^2 \leq (\|\mathcal{Y} - \mathcal{X}_p\|_{s^*} + \lambda) \sum_{k=1}^{K} \|\boldsymbol{\Delta}_k'\|_* + (\|\mathcal{Y} - \mathcal{X}_p\|_{s^*} - \lambda) \sum_{k=1}^{K} \|\boldsymbol{\Delta}_k''\|_*. \qquad (18)$$

Since

$$\|\mathcal{Y} - \mathcal{X}_p\|_{s^*} \leq \sigma \|\mathcal{E}\|_{s^*} + \|\mathcal{X}^* - \mathcal{X}_p\|_{s^*},$$

if $\lambda \geq \sigma \|\mathcal{E}\|_{s^*} + \|\mathcal{X}^* - \mathcal{X}_p\|_{s^*}$, the second term in the right-hand side of inequality (18) can be ignored and we have

$$\frac{1}{2}\|\mathcal{X}_p - \hat{\mathcal{X}}\|_F^2 \leq 2\lambda \sum_{k=1}^{K} \|\boldsymbol{\Delta}_k'\|_*$$
$$\leq 2\lambda \sum_{k=1}^{K} \sqrt{2r_k}\|\boldsymbol{\Delta}_k'\|_F$$
$$\leq 2\lambda \sum_{k=1}^{K} \sqrt{2r_k}\|\boldsymbol{\Delta}_k\|_F$$
$$\leq 2\sqrt{2}\lambda \sqrt{\sum_{k=1}^{K} r_k} \sqrt{\sum_{k=1}^{K} \|\boldsymbol{\Delta}_k\|_F^2}, \qquad (19)$$

where in the second line we used a simple observation that $\text{rank}(\boldsymbol{\Delta}_k') \le 2r_k$.

Next, we relate the norm $\|\mathcal{X}_p - \hat{\mathcal{X}}\|_F$ to the sum $\sum_{k=1}^K \|\boldsymbol{\Delta}_k\|_F^2$ in the right-hand side of inequality (19).

First suppose that $\sum_{k=1}^K \|\boldsymbol{\Delta}_k\|_F^2 \le \|\mathcal{X}_p - \hat{\mathcal{X}}\|_F^2$. Then from inequality (19), we have

$$\|\mathcal{X}_p - \hat{\mathcal{X}}\|_F \le 4\sqrt{2}\lambda \sqrt{\sum_{k=1}^K r_k}$$

by dividing both sides by $\|\mathcal{X}_p - \hat{\mathcal{X}}\|_F$.

On the other hand, if $\|\mathcal{X}_p - \hat{\mathcal{X}}\|_F^2 \le \sum_{k=1}^K \|\boldsymbol{\Delta}_k\|_F^2$, we use the following lemma

**Lemma 4.** *Suppose* $\{\boldsymbol{M}_p^{(k)}\}_{k=1}^K, \{\hat{\boldsymbol{M}}^{(k)}\}_{k=1}^K \in \mathcal{M}(\rho)$, *and* $\boldsymbol{S}^{(k)}$ *is constructed as a Kronecker product of* $K-1$ *ortho-normal matrices* $\hat{\boldsymbol{P}}^{(\ell)}$ *as* $\boldsymbol{S}^{(k)} = \hat{\boldsymbol{P}}^{(k-1)} \otimes \cdots \otimes \hat{\boldsymbol{P}}^{(k+1)}$, *where* $(\hat{\boldsymbol{P}}^{(\ell)})^\top \hat{\boldsymbol{P}}^{(\ell)} = \boldsymbol{I}_H$ *for* $\ell = 1, \ldots, K$. *Then for* $\mathcal{X}_p = \sum_{k=1}^K \text{fold}_k\left(\boldsymbol{M}_p^{(k)} \boldsymbol{S}^{(k)\top}\right)$ *and* $\hat{\mathcal{X}} = \sum_{k=1}^K \text{fold}_k\left(\hat{\boldsymbol{M}}^{(k)} \boldsymbol{S}^{(k)\top}\right)$, *the following inequality holds:*

$$\frac{1}{2}\sum_{k=1}^K \|\boldsymbol{\Delta}_k\|_F^2 \le \frac{1}{2}\|\mathcal{X}_p - \hat{\mathcal{X}}\|_F^2 + \rho \max_k(\sqrt{n_k} + \sqrt{H^{K-1}}) \sum_{k=1}^K \|\boldsymbol{\Delta}_k\|_*. \tag{20}$$

*Proof.* The proof is presented in Section D.5. □

Combining inequalities (18) and (20), we have

$$\frac{1}{2}\sum_{k=1}^K \|\boldsymbol{\Delta}_k\|_F^2 \le \left(\|\mathcal{Y} - \mathcal{X}_p\|_{s^*} + \rho \max_k(\sqrt{n_k} + \sqrt{H^{K-1}}) + \lambda\right) \sum_{k=1}^K \|\boldsymbol{\Delta}_k'\|_*$$
$$+ \left(\|\mathcal{Y} - \mathcal{X}_p\|_{s^*} + \rho \max_k(\sqrt{n_k} + \sqrt{H^{K-1}}) - \lambda\right) \sum_{k=1}^K \|\boldsymbol{\Delta}_k''\|_*.$$

Thus if we take $\lambda \ge \sigma\|\mathcal{E}\|_{s^*} + \|\mathcal{X}^* - \mathcal{X}_p\|_{s^*} + \rho \max_k(\sqrt{n_k} + \sqrt{H^{K-1}})$, the second term in the right-hand side can be ignored and following the derivation leading to inequality (19) and dividing both sides by $\sqrt{\sum_{k=1}^K \|\boldsymbol{\Delta}_k\|_F^2}$, we have

$$\|\mathcal{X}_p - \hat{\mathcal{X}}\|_F \le \sqrt{\sum_{k=1}^K \|\boldsymbol{\Delta}_k\|_F^2} \le 4\sqrt{2}\lambda \sqrt{\sum_{k=1}^K r_k},$$

where the first inequality follows from the assumption.

The final step of the proof is to bound the norm $\|\mathcal{E}\|_{s^*}$ with sufficiently high probability. By Lemma 1,

$$\|\mathcal{E}\|_{s^*} = \max_k \|\boldsymbol{E}_{(k)}\boldsymbol{S}^{(k)}\|.$$

Therefore, taking the union bound, we have

$$\mathbb{P}\left(\max_k \|\boldsymbol{E}_{(k)}\boldsymbol{S}^{(k)}\| \ge t\right) \le \sum_{k=1}^K \mathbb{P}\left(\|\boldsymbol{E}_{(k)}\boldsymbol{S}^{(k)}\| \ge t\right). \tag{21}$$

Now since each $\boldsymbol{E}_{(k)}\boldsymbol{S}^{(k)} \in \mathbb{R}^{n_k \times H^{K-1}}$ is a random matrix with i.i.d. standard Gaussian entries,

$$\mathbb{P}\left(\|\boldsymbol{E}_{(k)}\boldsymbol{S}^{(k)}\| \ge \sqrt{n_k} + \sqrt{H^{K-1}} + t\right) \le \exp(-t^2/(2\sigma^2)).$$

Therefore, choosing $t = \max_k(\sqrt{n_k} + \sqrt{H^{K-1}}) + \sqrt{2\log(K/\delta)}$ in inequality (21), we have

$$\max_k \|\boldsymbol{E}_{(k)}\boldsymbol{S}^{(k)}\| \le \max_k(\sqrt{n_k} + \sqrt{H^{K-1}}) + \sqrt{2\log(K/\delta)},$$

with probability at least $1 - \delta$. Plugging this into the condition for the regularization parameter $\lambda$, we obtain what we wanted.

### D.4  Proof of Lemma 2

*Proof.*  i) Let $\otimes_{k'\in[K]\backslash k}\boldsymbol{U}^{(k')}$ denote $\boldsymbol{U}^{(1)} \otimes \cdots \otimes \boldsymbol{U}^{(k-1)} \otimes \boldsymbol{U}^{(k+1)} \otimes \cdots \otimes \boldsymbol{U}^{(K)}$. We have

$$
\begin{aligned}
\boldsymbol{X}^*_{(k)} &= \boldsymbol{U}^{(k)}\boldsymbol{C}_{(k)}\left(\otimes_{k'\in[K]\backslash k}\boldsymbol{U}^{(k')}\right)^\top \\
&= \boldsymbol{U}^{(k)}\boldsymbol{C}_{(k)}\left(\otimes_{k'\in[K]\backslash k}(\boldsymbol{U}^{(k')})^\top\right) \\
&= \boldsymbol{P}^{(k)}\boldsymbol{\Lambda}^{(k)}(\boldsymbol{Q}^{(k)})^\top.
\end{aligned}
$$

Because of the minimality of the Tucker decomposition (5), $\boldsymbol{X}^*_{(k)}$, $\boldsymbol{C}_{(k)}$ and $\boldsymbol{U}^{(k)}$ are all of rank $R$, for all $k \in [K]$. Therefore, both $\boldsymbol{C}_{(k)}$ and $\otimes_{k'\in[K]\backslash k}(\boldsymbol{U}^{(k')})^\top$ have full row rank.

Hence, $\boldsymbol{C}_{(k)}$ has a Moore-Penrose pseudo inverse $\boldsymbol{C}^\dagger_{(k)}$ such that $\boldsymbol{C}_{(k)}\boldsymbol{C}^\dagger_{(k)} = \boldsymbol{I}$, and so does $\otimes_{k'\in[K]\backslash k}(\boldsymbol{U}^{(k')})^\top$. As a result, we have

$$\boldsymbol{U}^{(k)} = \boldsymbol{P}^{(k)}\boldsymbol{\Lambda}^{(k)}(\boldsymbol{Q}^{(k)})^\top \left(\otimes_{k'\in[K]\backslash k}(\boldsymbol{U}^{(k')})^\top\right)^\dagger \boldsymbol{C}^\dagger_{(k)}.$$

ii) Similarly, we have

$$\boldsymbol{Q}^{(k)}\boldsymbol{\Lambda}^{(k)}(\boldsymbol{P}^{(k)})^\top = \left(\otimes_{k'\in[K]\backslash k}\boldsymbol{U}^{(k')}\right)\boldsymbol{C}^\top_{(k)}(\boldsymbol{U}^{(k)})^\top.$$

By the definition of SVD, $\boldsymbol{\Lambda}$ is invertible and $(\boldsymbol{P}^{(k)})^\top\boldsymbol{P}^{(k)} = \boldsymbol{I}$. Hence,

$$\boldsymbol{Q}^{(k)} = \left(\otimes_{k'\in[K]\backslash k}\boldsymbol{U}^{(k')}\right)\boldsymbol{C}^\top_{(k)}(\boldsymbol{U}^{(k)})^\top\boldsymbol{P}^{(k)}(\boldsymbol{\Lambda}^{(k)})^{-1}.$$

This means $\boldsymbol{Q}^{(k)} \in \text{span}\left(\otimes_{k'\in[K]\backslash k}\boldsymbol{U}^{(k')}\right)$ and we then conclude $\boldsymbol{Q}^{(k)} \in \text{span}\left(\otimes_{k'\in[K]\backslash k}\boldsymbol{P}^{(k')}\right)$ using (i).

$\square$

## D.5 Proof of Lemma 4

Expanding $\mathcal{X}_p$ and $\hat{\mathcal{X}}$, we have

$$
\begin{aligned}
\|\mathcal{X}_p - \hat{\mathcal{X}}\|_F^2 &= \|\textstyle\sum_{k=1}^K \mathrm{fold}_k\big(\boldsymbol{\Delta}_k \boldsymbol{S}^{(k)\top}\big)\|_F^2 \\
&= \sum_{k=1}^K \|\boldsymbol{\Delta}_k\|_F^2 + \sum_{k \neq \ell} \langle \mathrm{fold}_k(\boldsymbol{\Delta}_k \boldsymbol{S}^{(k)\top}), \mathrm{fold}_\ell(\boldsymbol{\Delta}_\ell \boldsymbol{S}^{(\ell)\top}) \rangle \\
&= \sum_{k=1}^K \|\boldsymbol{\Delta}_k\|_F^2 + \sum_{k \neq \ell} \langle \mathrm{fold}_k(\boldsymbol{\Delta}_k) \times_{k' \neq k} \widehat{\boldsymbol{P}}^{(k')}, \mathrm{fold}_\ell(\boldsymbol{\Delta}_\ell) \times_{\ell' \neq \ell} \widehat{\boldsymbol{P}}^{(\ell')} \rangle \\
&= \sum_{k=1}^K \|\boldsymbol{\Delta}_k\|_F^2 + \sum_{k \neq \ell} \langle \mathrm{fold}_k(\boldsymbol{\Delta}_k) \times_\ell \widehat{\boldsymbol{P}}^{(\ell)}, \mathrm{fold}_\ell(\boldsymbol{\Delta}_\ell) \times_k \widehat{\boldsymbol{P}}^{(k)} \rangle \\
&= \sum_{k=1}^K \|\boldsymbol{\Delta}_k\|_F^2 - \sum_{k \neq \ell} \langle \boldsymbol{\Delta}_k (\boldsymbol{I}_H \otimes \cdots \otimes \widehat{\boldsymbol{P}}^{(\ell)} \otimes \cdots \otimes \boldsymbol{I}_H)^\top, \widehat{\boldsymbol{P}}^{(k)} (\mathrm{fold}_\ell(\boldsymbol{\Delta}_\ell))_{(k)} \rangle \\
&\geq \sum_{k=1}^K \|\boldsymbol{\Delta}_k\|_F^2 - \sum_{k \neq \ell} \|\boldsymbol{\Delta}_k\|_* \cdot \|(\mathrm{fold}_\ell(\boldsymbol{\Delta}_\ell))_{(k)}\| \\
&\geq \sum_{k=1}^K \|\boldsymbol{\Delta}_k\|_F^2 - 2\rho \max_k (\sqrt{n_k} + \sqrt{H^{K-1}}) \sum_{k=1}^K \|\boldsymbol{\Delta}_k\|_*,
\end{aligned}
$$

from which the lemma holds. Here we regarded $\mathrm{fold}_k(\boldsymbol{\Delta}_k \boldsymbol{S}^{(k)})$ as a Tucker decomposition with the core tensor $\mathrm{fold}_k(\boldsymbol{\Delta}_k)$ and factor matrices $\widehat{\boldsymbol{P}}^{(k')}$ for $k' \neq k$. Most of the factors except for $k$ and $\ell$ cancel out when calculating the inner product between two such tensors in the third line, because $(\widehat{\boldsymbol{P}}^{(k')})^\top \widehat{\boldsymbol{P}}^{(k')} = \boldsymbol{I}_H$. After unfolding the inner product at the $k$th mode in the fifth line, we notice that a multiplication by an ortho-normal matrix does not affect the nuclear norm or the spectral norm. In the last line we used $\{\boldsymbol{\Delta}_k\}_{k=1}^K \in \mathcal{M}(2\rho)$, which follows from the assumption that both $\{\boldsymbol{M}_p^{(k)}\}_{k=1}^K, \{\hat{\boldsymbol{M}}^{(k)}\}_{k=1}^K \in \mathcal{M}(\rho)$. $\qquad\square$

# E   Generalization of Theorem 1 to the higher rank case

**Theorem 4.** *Suppose that $X = \sum_{r=1}^{R} \beta_r u_r v_r^\top$, where $u_1, \ldots, u_R \in \mathbb{R}^n$ and $v_1, \ldots, v_R \in \mathbb{R}^m$ are unit orthogonal vectors respectively. Let $\tilde{X} = X + \sigma E$ be the noisy observation of $X$. There exists an universal constant $C$ such that with probability at least $1 - 3e^{-n}$, if $m/n \geq C(\beta_1/\beta_R)^4$, then*

$$
|\cos(\hat{U}, U)| \geq \begin{cases} 1 - \dfrac{Cmn}{(\beta_R/\sigma)^4}, & if \quad \dfrac{\beta_1}{\sqrt{m}} < \sigma \leq \dfrac{\beta_R}{(Cmn)^{\frac{1}{4}}}, \\[2ex] 1 - \dfrac{Cn(\beta_1/\beta_R)^2}{(\beta_R/\sigma)^2}, & if \quad \sigma \leq \dfrac{\beta_1}{\sqrt{m}}, \end{cases}
$$

*otherwise,* $|\cos(\hat{U}, U)| \geq 1 - \dfrac{Cn(\beta_1/\beta_R)^2}{(\beta_R/\sigma)^2}$ *if* $\sigma \leq \beta_R^2/(Cn)^{\frac{1}{2}}\beta_1$.

Suppose that $X = \sum_{r=1}^{R} \beta_r^2 u_r v_r^\top$ and $\tilde{X} = X + \sigma E$. We consider the second moment of $\tilde{X}$:

$$
\tilde{X}\tilde{X}^\top = \sum_{r=1}^{R} \beta_r^2 u_r u_r^\top + \sigma \left( \sum_{r=1}^{R} \beta_r \left( u_r v_r^\top E^\top + E v_r u_r^\top \right) \right) + \sigma^2 E E^\top
$$

$$
= \underbrace{\sum_{r=1}^{R} \beta_r^2 u_r u_r^\top + m\sigma^2 I}_{B} +
$$

$$
\underbrace{\sigma^2 E E^\top - m\sigma^2 I + \sigma \left( \sum_{i=1}^{R} \beta_r \left( u_r v_r^\top E^\top + E v_r u_r^\top \right) \right)}_{G}.
$$

The eigenvalue decomposition of $B$ can be written as

$$
B = [U \; U_2] \begin{bmatrix} \Sigma + m\sigma^2 I & \\ & m\sigma^2 I \end{bmatrix} \begin{bmatrix} U^\top \\ U_2^\top \end{bmatrix},
$$

where $U \in \mathbb{R}^{n \times R}$ and $\Sigma = \mathrm{diag}(\beta_1^2, \ldots, \beta_R^2)$. Similarly, the eigenvalue decomposition of $\tilde{X}\tilde{X}^\top$ can be written as

$$
\tilde{X}\tilde{X}^\top = [\hat{U} \; \hat{U}_2] \begin{bmatrix} \hat{\Sigma} & \\ & \hat{\Sigma}' \end{bmatrix} \begin{bmatrix} \hat{U}^\top \\ \hat{U}_2^\top \end{bmatrix},
$$

where $\hat{\Sigma} = \mathrm{diag}(\hat{\lambda}_1, \ldots, \hat{\lambda}_R)$ and $\hat{\Sigma}' = \mathrm{diag}(\hat{\lambda}_{R+1}, \ldots, \hat{\lambda}_n)$ s.t. $\hat{\lambda}_1 \geq \cdots \geq \hat{\lambda}_n$ are the eigenvalues of $\tilde{X}\tilde{X}^T$.

We first show a deterministic lower bound for $|\sin(\hat{U}, U)|$ assuming $\beta_R^2 \geq 2\|G\|$. Then we bound the spectral norm $\|G\|$ of the noise term (Lemma 3) and derive the sufficient condition for $\beta_R^2$.

The maximum singular value of $m\sigma^2 I$ is $m\sigma^2$. The minimum singular value of $\hat{\Sigma}$ is $|\hat{\lambda}_R|$. By Wely's theorem, $\|G\| \geq |\hat{\lambda}_R - \beta_R^2 - m\sigma^2|$, which means

$$
\hat{\lambda}_R \geq m\sigma^2 + \beta_R^2 - \|G\|.
$$

Let $R = G\hat{U}$. Since $\beta_R^2 \geq 2\|G\|$, we can apply the Wedin theorem and obtain

$$
|\sin(\hat{U}, U)| = \|U_2^\top \hat{U}\| \leq \frac{\|R\|}{\beta_R^2 - \|G\|} = \frac{\|G\hat{U}\|}{\beta_R^2 - \|G\|} \leq \frac{\|G\|}{\beta_R^2 - \|G\|} \leq \frac{2\|G\|}{\beta_R^2},
$$

where we used the property that the spectral norm is sub-multiplicative and $\|\hat{U}\| = 1$ in the second to last step.

Therefore,

$$| \cos(\hat{\boldsymbol{U}}, \boldsymbol{U})| \geq \sqrt{1 - \frac{4\|\boldsymbol{G}\|^2}{\beta_R^4}} \geq 1 - \frac{4\|\boldsymbol{G}\|^2}{\beta_R^4},$$

if $\beta_R^2 \geq 2\|\boldsymbol{G}\|$. It follows from Lemma 5 (shown below) that with probability at least $1 - 3e^{-n}$

$$\|\boldsymbol{G}\| \leq \begin{cases} 2\bar{C}\sigma^2\sqrt{mn}, & \text{if} \quad \beta_1/\sigma < \sqrt{m}, \\ 2\bar{C}\sigma\sqrt{n}\beta_1, & \text{otherwise}, \end{cases}$$

where $\bar{C}$ is an universal constant.

Let $C = 16\bar{C}^2$. Now consider the first situation where $m/n > C(\beta_1/\beta_R)^4$. If $\sigma > \frac{\beta_1}{\sqrt{m}}$, we have $\|\boldsymbol{G}\| \leq \frac{\sigma^2}{2}(Cmn)^{\frac{1}{2}}$. Meanwhile, if $\sigma \leq \frac{\beta_R}{(Cmn)^{\frac{1}{4}}}$, then we have $\beta_R^2 \geq \sigma^2(Cmn)^{\frac{1}{2}} \geq 2\|\boldsymbol{G}\|$. Combining these two conditions we obtain the first case in the theorem. When $\sigma \leq \frac{\beta_1}{\sqrt{m}}$, we can see that $\|\boldsymbol{G}\| \leq \frac{\sigma}{2}(Cn)^{\frac{1}{2}}\beta_1$. Moreover, since $m/n > C(\beta_1/\beta_R)^4$, it is implied that $\sigma \leq \beta_R^2/(Cn)^{\frac{1}{2}}\beta_1$ and thus $\beta_R^2 \geq 2\|\boldsymbol{G}\|$. This gives us the second case.

On the other hand, if $m/n \leq C(\beta_1/\beta_R)^4$, we require $\sigma \leq \beta_R^2/(Cn)^{\frac{1}{2}}\beta_1$ to obtain the last case in the theorem.

**Lemma 5.** *Let $\boldsymbol{G}$ be constructed as in the proof of Theorem 4. If $m \geq n$, there exists an universal constant $\bar{C}$ such that*

$$\|\boldsymbol{G}\| \leq \bar{C}\sigma^2\left(\sqrt{mn} + \sqrt{n}\beta_1/\sigma\right),$$

*with probability at least $1 - 3e^{-n}$.*

*Proof.* The proof is an $\varepsilon$-net argument. Let

$$\lambda = 2\sigma^2\left(\sqrt{4mn} + 4n + \sqrt{R + 8n + 4\sqrt{Rn}} \cdot \beta_1/\sigma\right).$$

The goal is to control $|\boldsymbol{x}^\top \boldsymbol{G} \boldsymbol{x}|$ for all the vectors $\boldsymbol{x}$ on the unit Euclidean sphere $\mathcal{S}^{n-1}$. In order to do this, we first bound the probability of the tail event $|\boldsymbol{x}^\top \boldsymbol{G} \boldsymbol{x}| > \lambda$, for any fixed $\boldsymbol{x} \in \mathcal{S}^{n-1}$. Then we bound the probability that $|\boldsymbol{x}^\top \boldsymbol{G} \boldsymbol{x}| > \lambda$ for all the vectors in a $\varepsilon$-net $\mathcal{N}_\varepsilon$. Finally, we establish the connection between $\sup_{\boldsymbol{x} \in \mathcal{N}_\varepsilon} |\boldsymbol{x}^\top \boldsymbol{G} \boldsymbol{x}|$ and $\|\boldsymbol{G}\|$.

To bound $\mathbb{P}(|\boldsymbol{x}^\top \boldsymbol{G} \boldsymbol{x}| > \lambda)$ for a fix $\boldsymbol{x} \in \mathcal{S}^{n-1}$, we expand $\boldsymbol{x}^\top \boldsymbol{G} \boldsymbol{x}$ as

$$\boldsymbol{x}^\top \boldsymbol{G} \boldsymbol{x} = \sigma^2(\|\boldsymbol{z}\|^2 - m) + 2\sigma\sum_{r=1}^{R}\beta_r\gamma_r(\boldsymbol{u}_r^\top \boldsymbol{x}),$$

where $\boldsymbol{z} = \boldsymbol{E}^\top \boldsymbol{x}$ and $\gamma_r = \boldsymbol{v}_r^\top \boldsymbol{z}$. It is easy to see that $\gamma_r \sim \mathcal{N}(0, 1)$, $\boldsymbol{z} \sim \mathcal{N}(0, \boldsymbol{I})$ and $\|\boldsymbol{z}\|^2$ is $\chi^2$ distributed with $m$ degrees of freedom.

Let $\boldsymbol{\gamma} = [\gamma_1, \ldots, \gamma_R]$ and $\boldsymbol{\omega} = [\boldsymbol{u}_1^\top x, \ldots, \boldsymbol{u}_R^\top x]$. We have

$$\begin{aligned} |\boldsymbol{x}^\top \boldsymbol{G} \boldsymbol{x}| &\leq \sigma^2\big|\|\boldsymbol{z}\|^2 - m\big| + 2\sigma\left|\sum_{r=1}^{R}\beta_r\gamma_r(\boldsymbol{u}_r^\top \boldsymbol{x})\right| \\ &\leq \sigma^2\big|\|\boldsymbol{z}\|^2 - m\big| + 2\sigma\sum_{r=1}^{R}\max_{r\in[R]}|\beta_r|\cdot|\gamma_r|\cdot|\boldsymbol{u}_r^\top \boldsymbol{x}| \\ &\leq \sigma^2\big|\|\boldsymbol{z}\|^2 - m\big| + 2\sigma\beta_1\cdot\|\boldsymbol{\gamma}\|\cdot\|\boldsymbol{\omega}\| \\ &\leq \sigma^2\big|\|\boldsymbol{z}\|^2 - m\big| + 2\sigma\beta_1\cdot\|\boldsymbol{\gamma}\|, \end{aligned}$$

where we used the Cauchy-Schwarz inequality in the second to last line and the fact $\|\boldsymbol{\omega}\| \leq 1$ in the last line. Note that $\gamma_1, \ldots, \gamma_R$ are i.i.d standard Gaussian distributed so that $\|\boldsymbol{\gamma}\|^2$ is $\chi^2$ distributed with $R$ degrees.

First we bound the deviation of the $\chi_m^2$ term. By the corollary of Lemma 1 in [17], we have

$$\mathbb{P}\big(\big|\|\boldsymbol{z}\|^2 - m\big| > \lambda_1\big) \leq 2e^{-4n}, \tag{22}$$

where $\lambda_1 = 2(\sqrt{4mn} + 4n)$.

Next we bound the $\chi_R^2$ term. Similarly, we have

$$\mathbb{P}(\|\boldsymbol{\gamma}\|^2 - R > \lambda_2) \leq e^{-4n}, \tag{23}$$

where $\lambda_2 = 2(\sqrt{4Rn} + 4n)$.

Combining inequalities (22) and (23), we have

$$\begin{aligned}
&\mathbb{P}(|\boldsymbol{x}^\top \boldsymbol{G} \boldsymbol{x}| > \lambda) \\
&\leq \mathbb{P}\left(\sigma^2 \big|\|\boldsymbol{z}\|^2 - m\big| + 2\sigma\beta_1\|\boldsymbol{\gamma}\| > \sigma^2\lambda_1 + 2\sigma\beta_1\sqrt{R+\lambda_2}\right) \\
&\leq \mathbb{P}\left(\big|\|\boldsymbol{z}\|^2 - m\big| > \lambda_1 \vee \|\boldsymbol{\gamma}\| > \sqrt{R+\lambda_2}\right) \\
&\leq \mathbb{P}\left(\big|\|\boldsymbol{z}\|^2 - m\big| > \lambda_1\right) + \mathbb{P}\left(\|\boldsymbol{\gamma}\| > \sqrt{R+\lambda_2}\right) \\
&\leq 3e^{-4n},
\end{aligned}$$

where the second to last line follows from the union bound.

Furthermore, using Lemma 5.2 and 5.4 of [27], for any $\varepsilon \in [0,1)$, it holds that

$$|\mathcal{N}_\varepsilon| \leq (1 + 2/\varepsilon)^n,$$

and

$$\|\boldsymbol{G}\| \leq (1 - 2\varepsilon)^{-1} \sup_{\boldsymbol{x} \in \mathcal{N}_\varepsilon} |\boldsymbol{x}^\top \boldsymbol{G} \boldsymbol{x}|.$$

Taking the union bound over all the vectors in $\mathcal{N}_{1/4}$, we obtain

$$\mathbb{P}(\|\boldsymbol{G}\| \leq 2\lambda) \leq \mathbb{P}\left(\sup_{\boldsymbol{x} \in \mathcal{N}_{1/4}} |\boldsymbol{x}^\top \boldsymbol{G} \boldsymbol{x}| > \lambda\right) \leq |\mathcal{N}_{1/4}| 3e^{-4n} < 3e^{-n}.$$

Finally, the statement is obtained by noticing that $n \leq m$ and $R \leq n$. $\qquad\square$