[Reviews · NeurIPS 2015]

Submitted by Assigned_Reviewer_1

This paper presents a subspace norm minimization solution for low rank tensor denoising problem. Theoretically analysis of the recovery guarantee is also provided. The paper is relatively easy to read and the technique part is sound. The drawback is the experiments and the applicability of the algorithm. Please see the detailed comments below.

1. Please briefly introduce the problem and the notations in the Introduction section, which will make the paper easier to follow.

2. The analysis and the proposed algorithm seems to tackle the low CP rank tensor completion problem. However, at the beginning of Section 3, the authors mention the low multilinear rank of a tensor. Is the proposed method applicable for Tucker form tensor completion?

3. In the subspace norm minimization, the author use a direct summation of the trace norm of M^(k). Why not use a weighted summation?

4. The main concern of the proposed algorithm is the computational complexity, as it is a main difficulty in solving the tensor completion problem. Both step (ii) and step (iii) in the algorithm are time consuming. The authors are suggested to add more analysis for the computational complexity. And it is better to show the running times of all the competing algorithms in the experiments. Currently, only one small scale real data is included in the experiments. More real data is preferred to be used.

5. Could the authors give some comments on the situation that the relative error bigger than 1 in Figure 2? This is not a normal case.

6. The recovery guarantee relies on the incoherence of matrix M^(k). Could the authors show some examples which satisfy this condition? What will the algorithm derive if the condition is slightly violated?
Summary: This paper presents a subspace norm minimization solution for low rank tensor denoising problem. Theoretically analysis of the recovery guarantee is also provided. The paper is relatively easy to read and the technique part is sound. The drawback is the experiments and the applicability of the algorithm.

Submitted by Assigned_Reviewer_2

The authors deal with the problem of tensor recovery from noisy measurements. Their main advances are two:

1. Theoretically they prove that a signal to noise ratio of the order O(n^(K/4)) is sufficient for stable recovery, settling a recent conjecture [18]. This is done with an elegant and straightforward analysis for matrices, and extends to tensors through unfolding arguments.

2. They introduce a new norm, called subspace norm, which is constructed by all the truncated SVDs of the unfolded matrices along each dimension. The tensor to be found is modeled as a mixture of low rank tensors that accept low rank factorizations.

Quality: The paper is technically sound, with proofs for the theoretical arguments and description of the construction and computation of the subspace norm. The claims are supported by simulations.

Clarity: The paper is mostly clearly written. Section 2 is very clearly written. Section 3 is highly technical. Although written in a self-contained way, the average reader would probably benefit from an example of the subspace norm and the importance of the H parameter (number of singular vectors kept in the partial SVDs).

Also, the CP acronym for the method that the authors compare their method against is never defined. This makes it harder for a non-expert (like myself) to judge the relative performance of the method.

I think the discussion on how the choice of H blends together convex and non-convex approaches could be better motivated and the extreme cases (H=1, H=N) could be presented in an intuitive way.

Originality: The approach is original to the best of my knowledge.

Significance: The theoretical results may feel incremental but I think it's valuable because they settle an actively studied conjecture in a simple way that may be of usage beyond this specific problem.
Summary: A good paper that improves on the bounds for tensor completion and also comes with a novel competing algorithm. Due it's nature the paper is heavy on notation and can be hard to follow. A more elaborate presentation of the subspace norm would benefit the non-expert reader.

Submitted by Assigned_Reviewer_3

This paper introduces a new theoretical bound for recovering a low-rank tensor from a noisy observation as well as a new norm on tensors.

Among the neatest results in this paper is a demonstration and a proof that there is a two-phased behavior of the recovery.

There are a number of problems with the paper, but quality, clarity and significance are my main concerns.

The papers quality suffers mainly due to poor grammar, poor terminology, notation and terms that aren't properly and clearly defined.

It is unclear how significant any of the results are as the authors have not demonstrated a meaningful improvement in a practical setting.

The sub-space norm for example improves things when the relative error is close to 1, but why do I care what happens in a situation where the model is anyways useless?

I would have expected a discussion on the practical value of the result - especially given that the signal to noise ration required is quite large (even if reduced dramatically from earlier papers).

I can only presume that a lack of such a discussion means the value is marginal and that this should be seen as mainly a theoretical contribution?

Here are some comments

"Tensor is a natural way" --> "The tensor is a natural way" "for variety of data" --> "for a variety of data" "ranging from ..., neuroimaging" --> "ranging FROM ... TO neuroimaging" what is model 2 in the comment of table 1? "(Vershynin, 2010)" --> this should be a proper reference.

This is one of many demonstrations that the paper was not really completed.

"mn" vs "nm" : Just be consistent with the order of the product so as not to distract the reader with irrelevant details.

Definition (4) is completely bogus.

You can't redefine "k-1, ..., k+1" to mean "1,...,k-1,k+1,...,K".

Also, you only sometimes use this definition.

Even the right hand of (4) is written in a terrible way.

In stead of writing

A^{(k-1)}\otimes \cdots \otimes A^{(1)} \otimes A^{(K)} \otimes \cdots \otimes A^{(k+1)} I would prefer A^{(1)}\otimes \cdots \otimes A^{(k-1)} \otimes A^{(k+1)} \otimes \cdots \otimes A^{(K)} which clearly shows that the term A^{(k)} is the one left out.

Even clearer and more succinct would be \otimes_{i\neq k} A^{(i)}

"n_1\times \cdots n_K" should be "n_1\times \cdots \times n_K".

This sort of mistake is made repeatedly in the paper.

M is not properly defined.

I would in the least expect to see the sizes of M defined. Is there a missing sum in the definition of Z_{(k)}^{(k)}?

If not then you may as well say that Z has a rank one factorization as opposed to a low-rank factorization?

In any case things it's not exactly clear to me what exactly Z is.

"r=1,\ldots R" --> "r=1,\ldots,R"

All the graphs are too busy (and extremely small) to make sense of.

The experiments use a cheating type of method to find lambda, and the rank is assumed known.

I assume that in reality both lambda and the rank will be unknown.

The only real data experiment was the amino acids data example.

It's unclear that the paper added any practical value to analyzing these sort of data sets.

Summary: The paper gives some interesting new theoretical results, but fails to address several practical concerns and could be written much better.

Author Feedback
Author rebuttal: We thank the reviewers for their helpful comments, and respond briefly to the main comments and criticisms. All other suggestions will be reflected in the final version.

Reviewer 1:

CP (canonical polyadic) decomposition vs Tucker decomposition: The proposed method is also applicable for Tucker decomposition. Please note that CP decomposition is a special case of Tucker decomposition and our argument in Lemma 2 is valid for Tucker decomposition.

Weighted sum: We did not introduce weights because the weights can be subsumed in the choice of matrices S^(k).

Computational complexity: We'd like to point out that step (ii) and (iii) can be carried out efficiently since H (#left singular vectors we keep) is very small (<< n_k). Ahat^(k) is only n_k by H, and M^(k) is only n_k by H^(k-1). Comparing to existing convex methods e.g. latent trace norm [21], which is essentially the case H = n_k, our method is much faster due to the restriction to the subspace -- we view this as a strength of our paper.

Relative error larger than 1: we agree with the reviewer that the part with relative error above 1 should be ignored. Our claim here is that the proposed method is comparable to CP, for which no theoretical guarantee is available, and it significantly outperforms previously proposed methods with performance guarantees (latent /overlap). We'd like to refer the reviewer to Appendix C for results of L2 regularized CP.

Incoherence assumption: It is possible to choose a large rho such that the assumption is trivially satisfied. However, this results in a large lambda (see line 841 in Appendix) and will make the bound loose.

Reviewer 2:

CP stands for canonical polyadic (decomposition), which is defined in Eq. (5). CP in the experiments refers to an implementation of an algorithm to find an (approximate) CP decomposition implemented in Tensorlab [20].

Intuition for H: In the extreme case H=1, M^(k) is a vector and the model can be viewed as a leave-one-out HOSVD (nonconvex); i.e., every factor except the kth one is estimated in the first step and fixed. On the other hand, H=n recovers the latent trace norm ([21], convex).

Reviewer 3:

Notations:
- model (2) refers to equation (2).
- Definition 4 is valid and necessary because the kronecker product is non-commutative; in general A \otimes B is not equal to B \otimes A.
- Z^(k)_(k) is the the mode-k unfolding of tensor Z^(k). This notation is defined in line 76 and footnote 1 (page 2). When we define S^(k) as in (4), line 248 can be equivalently written as
Z^(k) = fold_k(M^(k)) \times_{1} A^(1) ... \times_{k-1} A^(k-1) \times_{k+1} A^(k+1) ... \times_K A^(K),
where A \times_k B is the k-mode product of a tensor A with a matrix B (see [13]).
- Both M^(k) and S^(k) are matrices with H^(k-1) columns. Thus there is no sum missing in line 248. The size of M^(k) is n_k by H^(k-1), which can be deduced from the sizes of Z^(k)_(k) and S^(k). We will make this explicit in the final version.

The choice of lambda and the rank: The choice of lambda is not cheating. In Theorem 2 we have fully specified how the regularization parameter lambda should scale to obtain the theoretical guarantee. We have shown through the experiments that this theoretically motivated choice of lambda ("suggested") performs nearly as well as the best choice. This theoretically motivated choice also performs better than other norm-based methods (with the best lambda) and comparable to CP (with the correct rank). The choice of the rank affects both CP and the proposed method in a similar way and this is why we only provide the results for different ranks in Appendix C. The results show that both CP and the proposed method benefit from regularization when the rank is overspecified.

Comparison to CP: Our method performs comparably to CP, but it comes with a theoretical guarantee.

Reviewer 4:

Comparison to Tucker: Since CP is a special case of Tucker and Tucker has more parameters than CP, we don't expect Tucker to perform better than CP in the current setting. Note that the synthetic data is generated from a CP model. The amino data is known to be well approximable by CP.

Reviewer 7:

In this paper, we have settled a conjecture posed by Richard & Montanari [18], and showed that indeed O(n^{K/4}) signal-to-noise ratio is sufficient also for odd order tensors. Moreover, our analysis shows an interesting two-phase behavior of the error.

This finding lead us to the development of the proposed subspace norm. The proposed norm is defined with respect to a set of orthonormal matrices A^(1),...,A^(K), which are estimated by mode-wise singular value decompositions. We have analyzed the denoising performance of the proposed norm, and shown that the error can be bounded by the sum of two terms, which can be interpreted as an approximation error term coming from the first (non-convex) step, and an estimation error term coming from the second (convex) step.